# Most Random-Encounter-Model Density Estimates in Camera-Based Predator–Prey Studies Are Unreliable

**DOI:** 10.3390/ani14233361

**Published:** 2024-11-22

**Authors:** Sean M. Murphy, Benjamin S. Nolan, Felicia C. Chen, Kathleen M. Longshore, Matthew T. Simes, Gabrielle A. Berry, Todd C. Esque

**Affiliations:** U.S. Geological Survey, Western Ecological Research Center, Boulder City, NV 89005, USAfchen@usgs.gov (F.C.C.); longshore@usgs.gov (K.M.L.);

**Keywords:** abundance, *Canis latrans*, carnivore, ctmm, day range, *Lepus californicus*, Mojave Desert, Nevada, population density, unmarked

## Abstract

Population estimates are often required for identifying relationships between predators and their prey and to inform conservation and management actions. The random encounter model (REM) estimates population density of wildlife lacking individually unique markings, based on photographs or videos from remote camera-traps. However, the REM has strict sampling and input requirements that can be problematic, particularly for predators and other species which use landscapes non-randomly. Using data from a predator and its co-occurring prey, we found that placing cameras to target the predator, which may be implemented to achieve minimum sample sizes, inflated both predator and prey density estimates. Further, borrowing movement velocity (day range) values from other studies, species, or time periods caused substantial changes in density estimates. A comprehensive literature review revealed that 91% of REM density estimates in published predator–prey studies used data from non-random cameras or borrowed movement velocities and therefore did not satisfy REM requirements. Consequently, most REM density estimates from predator–prey ecology studies are likely not of the quality or reliability necessary for informing effective wildlife conservation or management.

## 1. Introduction

Relationships between predators and their prey at the population level remain a fundamental topic of investigation in population biology. Among terrestrial wildlife, perhaps the best known and most intensively studied predator–prey relationship is the approximately ten-year cycle of oscillations between co-occurring Canada lynx (*Lynx canadensis*) and snowshoe hare (*Lepus americanus*) populations in North America [1,2,3]. Identifying and quantifying such relationships is not only critical for understanding species ecology and population dynamics, but also for disentangling the myriad of ecological, climatic, and anthropogenic effects on population viability of both predator and prey populations [4,5]. Furthermore, successful recovery of many imperiled wildlife species, including via natural recolonization or reintroduction, can be strongly influenced by the abundance or density of predator or prey populations in recovery areas [6,7,8]. Thus, obtaining accurate and reliable estimates of abundance or density of both predator and prey populations is often critical to the identification, implementation, and effectiveness of conservation and management actions [9,10].

Although multiple methods exist for effectively surveying terrestrial wildlife populations to obtain detection data for estimating their abundance or density [11], globally, remote camera-trapping has become the most used non-invasive approach by wildlife researchers and managers [12,13,14]. Extensive adoption of camera-traps has occurred largely because of rapid technological advancements that facilitated improved monetary and logistical efficiency of the method, particularly for surveying rare or cryptic wildlife in remote or difficult-to-access locales [14,15,16]. Most analytical methods that are available for producing abundance or density estimates from photographic detections obtained at camera-traps require animals in a population to have unique physical markings that enable accurate individual identification (e.g., the rosettes on jaguars (*Panthera onca*) [17,18,19,20]). In contrast, most terrestrial wildlife species that are detectable with camera-traps do not have sufficiently unique natural physical characteristics that can be reliably used as ‘marks’ to determine an animal’s individual identity from photographs with high confidence [21,22,23]. Mark–resight approaches were developed to resolve this individual identifiability issue, whereby practitioners physically capture and artificially mark (e.g., with ear-tags or radio-collars) a portion of individuals in a population, and camera-trap detections of both marked and unmarked animals are integrated to produce reliable and robust population abundance and/or density estimates [24,25,26,27]. However, mark–resight methods can increase financial costs and logistical complexity of research studies and may not be applicable in some jurisdictions or to some species because of restrictions on physically capturing animals or applying artificial marks.

In lieu of capturing and artificially marking a portion of physically nondescript animals in a population, practitioners often apply methods to camera-trap detections of unmarked wildlife that produce indices (rather than estimates) of animal abundance or density [28,29]. Such indices have multiple issues and are typically unsuitable surrogates for empirical estimates in most conservation and management scenarios [13,30,31]. Thus, over the last two decades, considerable attention has been given to the development and application of analytical methods that can directly estimate population abundance or density from camera-trap surveys of unmarked wildlife, such as the spatial count (SC), unmarked spatial capture–recapture (uSCR), random encounter and staying time (REST), and camera-trap distance sampling (CTDS) models [32,33,34,35].

One of the first developed unmarked methods was the random encounter model (REM; [36]), which was founded on the theory of ideal gas particle collision (e.g., encounter or contact rate between animals and cameras) and estimates unmarked animal density within the collective viewsheds of camera-traps. The REM has three foundational assumptions, one of which is that animals move independently of camera-traps [36,37]. Satisfying that assumption requires that camera-traps be deployed randomly across a study area, such that specific habitats or landscape features that animals may preferentially select for or avoid (e.g., trails or roads) are not strategically or disproportionately sampled relative to the true presence of those features within a given study area [36,37]. The consequences of violating that assumption, e.g., by strategically placing cameras on animal travel routes, are that encounter rates can be artificially inflated, which should positively bias REM density estimates [38,39].

However, spatially random camera placement can be problematic for terrestrial predators, because most predators do not use landscapes randomly and are more likely to be detected by camera-traps placed on trails or other travel routes [39,40,41]. Consequently, random camera placement can result in insufficient detections of predators that precludes density estimation with the REM [36]. Multiple studies also have assumed that, if camera-traps are placed strategically to detect a predator, then those same cameras are conversely placed randomly relative to the prey species [42,43], but justification for that assumption is scant and recent research indicates it may be untenable [40,41]. For instance, detections of mountain hares (*Lepus timidus*) and roe deer (*Capreolus capreolus*) were 2–7× higher at camera-traps that were strategically placed to detect their predator, Eurasian lynx (*Lynx lynx*), than at randomly placed camera-traps [41]. Additionally, the underlying ideal gas particle collision framework of the REM necessarily invokes secondary data requirements to produce density estimates, including the need to obtain an estimate of animal movement velocity (day range) [36]. Methods exist to estimate movement velocities for the study species during the survey period from either camera-trapping data [44,45] or from satellite transmitter (e.g., global positioning systems [GPS] collar) tracking data that appropriately account for movement tortuosity [46,47], but researchers sometimes attempt neither approach and instead borrow velocity estimates from other study areas, time periods, or species [38,48,49]. However, space use of predators and prey, including movement velocity, often spatiotemporally varies within a species across its geographic range as a function of habitat, landscape, or climatic conditions; for example, jaguar movement velocity varies substantially among populations as a function of available forest cover [50]. Consequently, borrowing movement velocity values from other populations, study periods, or other species for use in the REM can severely bias density estimates [51,52], which may lead to flawed conservation and management decisions.

The use of camera-traps in predator–prey ecology studies, including research on population abundance- or density-based relationships, is expected to increase in the future [9,53]. Further, given that most terrestrial predators and their common prey species do not have individually unique natural markings, and considering the ease with which the REM can be applied to camera-trapping data of unmarked wildlife [48], it is reasonable to assume that the REM may be a considered approach in many future predator–prey ecology studies. To date, there have been few empirical tests of camera placement strategy and movement velocity sources on REM density estimates, and no assessments of the extent of these potential issues in previous predator–prey studies has been conducted. Therefore, using an empirical, multi-year predator–prey ecology dataset from continuous operation of both randomly and strategically placed camera-traps, as well as simultaneous GPS-collar-based tracking data from both the predator and prey species, we present an evaluation of the effects on REM density estimates of using detections from strategically placed cameras and borrowing movement velocity values. Additionally, we conducted a comprehensive, systematic literature review of studies that used the REM to estimate animal density, with a focus on predator–prey ecology studies, to investigate the extent and trends of these issues. Our findings have multiple implications for predator–prey ecology research, and conservation or management decisions based on studies that used, or intend to use, the REM to estimate population densities.

## 2. Materials and Methods

### 2.1. Predator–Prey System

Coyotes (*Canis latrans*) are terrestrial carnivores native to the historically non-forested deserts, prairies, and grasslands of the western United States, but have colonized much of North America over the last century [54,55]. In the desert Southwest, where coyotes are native, they can exert effects at multiple trophic levels, often varying by the presence or absence of larger carnivores in ecosystems. For instance, coyotes are typically subordinate mesopredators where gray wolves (*Canis lupus*) or pumas (*Puma concolor*) are present, but coyotes are often dominant apex predators where wolves and other large carnivores (e.g., brown bears [*Ursus arctos*]) are absent [56,57]. Thus, coyote dietary breadth can vary substantially among and within populations [57,58,59].

Throughout much of the desert Southwest, black-tailed jackrabbits (hereafter jackrabbits; *Lepus californicus*) are a dominant prey item in coyote diets [58,59,60]. Researchers have presumed that oscillations in jackrabbit populations are at least partially influenced by variation in predation pressure from coyote populations, which also presumably fluctuate in response to jackrabbit populations [61,62,63]. Increasing severity and duration of drought conditions and higher temperatures for longer periods in the desert Southwest, as well as recent outbreaks of lethal rabbit hemorrhagic disease virus 2 (RHDV2) across North America, have heightened concerns that jackrabbit populations are declining or may be locally extirpated in some areas [64,65]. Those effects could cause coyotes to ‘prey switch’ and depredate other species, some of which are imperiled (e.g., Mojave desert tortoise [*Gopherus agassizii*]; [60,66]). Therefore, monitoring both coyote and jackrabbit populations in the Southwest, including estimating seasonal and annual variation in their abundance and density, is critical for quantitatively evaluating this predator–prey relationship and identifying the potential conditions in which coyotes may switch to depredating on desert tortoises or other vulnerable wildlife.

### 2.2. Study Area

The study area was located in the east-central portion of the Mojave Desert, in southern Nevada, United States, and was centered on the Boulder City Conservation Easement (BCCE; Figure 1), which is a 353 km^2^ conservation area that was established in 1995 to protect habitats for the federally threatened Mojave desert tortoise and other fauna and flora of conservation importance [67]. Although the BCCE proper is protected from anthropogenic development and wildlife hunting, hunting is legal on surrounding lands. The study area comprises portions of three ecoregions: Eastern Mojave Basin, Eastern Mojave Low Ranges and Arid Footsteps, and Eastern Mojave Mountain Woodlands and Shrublands [68]. These ecoregions collectively form a complex landscape of rocky hills and mountains that envelop a high-elevation desert valley that is intersected by multiple drainages (washes), which are dry for most of a given year and interiorly drain toward the lowest spot in the valley. The dominant vegetation in upland areas is shrubs, primarily *Larrea tridentata* and *Ambrosia dumosa*, along with *Ephedra nevadensis*, *Thamnosma monatana*, and multiple species of cacti (e.g., *Cylindropuntia acanthocarpa*, *Opuntia basilaris*, etc.). At higher elevations in the surrounding mountains, *Yucca jaegeriana* are present in scattered stands. Multiple invasive plant species also occur in the area, including *Bromus rubens*, *Salsola* spp., and *Brassica tournefortii*. The climate is arid, with two distinct rainfall seasons, which generally encompass winter (November–April) and summer (May–October), respectively. Based on long-term climate monitoring data (1948–2021), mean winter and summer temperatures are 12 °C and 27 °C, respectively, and mean precipitation during winter and summer are 4.14 ± 6.08 cm (mean ± standard deviation [SD]) and 2.63 ± 4.50 cm, respectively [69].

### 2.3. Animal Capture and Tracking

During 2019–2021, we live-captured and collared both coyotes and jackrabbits throughout the study area. The resulting satellite tracking data were used to obtain movement velocity (day range) estimates for use in the REM [47,70]. All capture and handling methods were approved by a U.S. Geological Survey Animal Care and Use Committee (ACUC) and conformed to the recommendations of the American Society of Mammalogists for humane capture and handling of wild mammals in research [71].

To capture coyotes, we located sites that appeared to have been frequented by coyotes, deployed salvaged mule deer (*Odocoileus hemionus*) carcasses or commercially raised chicken as bait, and then set Minnesota Brand MB-550-RJ foothold traps (Minnesota Trapline Products, Inc.; Pennock, MN, United States). Coyote traps were opened during late afternoon or early evening, remained open until the following morning, and then were closed during the day to prevent heat-related illness or death during capture. We physically restrained captured coyotes using a Y-pole, after which we chemically immobilized coyotes using a syringe to administer ketamine and medetomidine at dosages of 2 mg/kg and 0.07 mg/kg, respectively [72]. We removed each anesthetized coyote from the trap, restrained it, and determined sex, weight, and age class (juvenile versus subadult and adult); age class was assigned based on tooth wear [73]. We outfitted each captured coyote with uniquely numbered vinyl ear tags and, on each subadult and adult that weighed >8 kg, a GPS collar (Lotek Litetrack Iridium 250 or 360, depending on body weight/size [Lotek Wireless, Inc., Newmarket, ON, Canada]). Coyote GPS collars were programmed to obtain satellite location fixes every three hours, except during the first five days of each month, during which collars obtained location fixes every hour. All coyote collars were equipped with electronic release mechanisms that were programmed to disconnect and cause the collar to fall off the coyote prior to the end of the GPS battery lifespan. After processing and collaring captured coyotes, we administered the antagonist atipamezole via syringe at a dosage of 0.35 mg/mg medetomidine for reversal [72].

To capture black-tailed jackrabbits, we located sites that appeared to have been frequented by jackrabbits, placed apples, hay, and/or carrots as bait at each site, and then deployed a cage trap (Tomahawk #109, 38 × 38 × 107 cm [Tomahawk Live Trap, Hazelhurst, WI, USA]; Comstock, 30 × 38 × 76 cm [Comstock Custom Cage, LLC, Gansevoort, NY, USA]). Jackrabbit traps were opened during late afternoon or early evening, remained open until they were checked before sunrise the following morning, and then were closed during the day to prevent heat-related illness or death during capture. When a jackrabbit was captured, we covered the trap with a blanket and removed the animal from the trap by placing a hand on the scruff of the neck with ears laid flat along the back and a second hand grasping the animal in front of the pelvis. We then restrained jackrabbits in a cloth bag or pillowcase and placed a hood, which incorporated a sock modified with ear plugs, over the head and face to limit visual and auditory stimuli. We determined sex, weight, and age class (juvenile versus subadult and adult) of each captured jackrabbit; age class was assigned based on body size and weight. We outfitted each captured jackrabbit with uniquely numbered vinyl ear tags and, on each subadult and adult that weighed >1.75 kg, a VHF radio-collar (Holohil; Holohil Systems, Carp, ON, Canada) with an attached i-gotU GPS logger (MobileAction, San Francisco, CA, USA) or a GPS collar (Sirtrack Litetrack RF30 [Lotek NZ, Havelock North, New Zealand]; Telonics TGW-4000-4 [Telonics, Inc., Mesa, AZ, USA]). The i-gotU GPS loggers were programmed to obtain satellite location fixes every 30 min; the Sirtrack GPS collars were programmed to obtain location fixes every three hours; and the Telonics GPS collars were programmed to obtain location fixes every four hours. All jackrabbit collars were equipped with release mechanisms that comprised cotton panels or latex tubing, which gradually deteriorated over time, causing the collars to fall off jackrabbits.

### 2.4. Movement Velocity Estimation

Similar to Henrich et al. [47], we estimated movement velocities for GPS-collared coyotes and jackrabbits during each season × year combination using continuous-time movement models (CTMM) that we implemented via the ctmm package (v1.2.0) in the R statistical computing environment (v4.3.0) [46,74,75,76,77]. Seasons were summer (1 May–31 October) and winter (1 November–30 April), based on long-term climate data from the study area [69], which reflected the ‘summer’ and ‘winter’ rainfall seasons in the Mojave Desert, respectively. We first subset the tracking location data for each animal by the season and year during which each location was obtained; we only used location data from an individual if a minimum of 100 and 200 locations were obtained during a season × year for jackrabbits and coyotes, respectively. Next, because three different GPS devices with different satellite location fix rates were deployed on jackrabbits, we thinned the location data to three- to four-hour fix intervals to prevent bias that could be caused by uneven sampling (irregular fix rates) among individuals. All coyote GPS collars had the same satellite location fix rates, so we did not thin the coyote tracking data. Then, for each individual within each season × year combination, we calculated empirical variograms that estimated the semi-variance as a function of time lag to visualize the temporal autocorrelation structure of location data [74,75]. For the coyote data, location fixes were obtained every hour during the first five days of each month, whereas fixes were obtained every three hours during the remainder of each month; therefore, we specified corresponding lag bin widths that were aggregated in the variogram calculations to account for the sampling interval changes [78].

To estimate movement velocities for both species, we fit integrated Ornstein–Uhlenbeck (IOU) and Ornstein–Uhlenbeck foraging (OUF) continuous-time movement models [74]. These models explicitly account for movement tortuosity and are not plagued by the problem of underestimating distance traveled by an animal between successive locations that other methods have, assuming the sampling frequency (fix rate interval) is not too coarse [46,79,80]. We fit both the correlated velocity IOU and OUF models, as well as the Ornstein–Uhlenbeck (OU) and Brownian motion (BM) models that excluded correlated velocities [46], and used Akaike’s Information Criterion corrected for small sample size (AIC*_c_*) for model selection [81]. If the top-ranked model for a given individual within a season × year was the OU or BM model, then the location data were too coarse to produce velocity estimates; in contrast, if the top-ranked model was the IOU or OUF model, then the data were of sufficient resolution to estimate velocity [46].

### 2.5. Camera-Trapping

We deployed two sets of camera-traps simultaneously to obtain photographic detections of coyotes and jackrabbits for estimating their population densities using the REM. The first set of camera-traps were deployed randomly on two separate grids, one in the northern portion of the study area (*n* = 20 cameras) and another in the southern portion of the study area (*n* = 20 cameras). In each grid, spacing among clusters of four sites was 1 km, whereas spacing among camera-traps within each cluster was 500 m. In both of those random sampling grids, we placed camera-traps at the centroid of each sampling cell and did not strategically target or avoid animal travel routes, habitat types, or landscape characteristics (i.e., completely random sampling). In contrast, we deployed the second set of camera-traps strategically (*n* = 43 cameras) on identified coyote travel routes. Site selection for the strategic cameras was based on physical observation of coyote sign (e.g., tracks or scat) and preliminary data from GPS collar telemetry monitoring of coyotes. Given the topography of the desert environment in the study area, coyotes frequently used drainages (washes) as travel corridors; thus, most of the strategically placed cameras were placed in drainages (i.e., non-random sampling). Although the randomly placed cameras remained at their respective locations for the duration of the survey (i.e., were not moved), we were forced to relocate some of the strategically placed cameras during the multi-year survey period because of vandalism/theft or weather conditions (e.g., flooding in the drainage the camera was originally placed in), resulting in the strategic sampling of 63 total unique locations with the 43 cameras.

All cameras (Bushnell [Overland Park, KS, United States] #119876, #119876C, or Moultrie [Birmingham, AL, United States] #M-40) were placed in metal lock boxes, secured to steel T-posts or natural shrubs approximately 30 cm above the ground, and were programmed to take photographs whenever movement was detected. Randomly placed cameras were programmed with a one-minute interval between successive bursts of six photographs, whereas strategically placed cameras were programmed with a 30 s interval between successive bursts of six photographs. However, prior to data analysis, photograph detections from the strategically placed cameras were subset to one-minute intervals for compatibility and standardization of detections between the two camera placement types. We operated cameras continuously from October 2019 to October 2021, and we checked each camera to download data and replace batteries every three to six months. We used the CameraSweet package of software (v6.1:2020.12.25; https://smallcats.org/camerasweet/) to organize and process photographs [82,83], and we grouped all coyote and jackrabbit detections by the season (winter versus summer) within the year that they occurred. To separate unique detections from likely successive detections of the same individual, we used 30 min and 10 min discrimination thresholds between photograph bursts for coyotes and jackrabbits, respectively [84,85,86]. Those discrimination thresholds were determined based on a preliminary analysis of camera detections of collared individuals of both species, using both the location data from GPS collars and the corresponding camera detections to evaluate the average time that each species spent in proximity to cameras and were potentially detectable.

### 2.6. REM Density Estimation

We used the REM to estimate seasonal densities of both coyotes and jackrabbits within each year. We considered all individual coyotes and jackrabbits as unmarked regardless of whether or not they had GPS collars or ear tags. The REM requires users to specify three camera-specific model parameters [36]: the radial detection distance from the camera (*r*), the zone (arc) of detection (θ), and the camera operation time (*t*; days). Because our objective was to evaluate the potential consequences of violating requirements of the REM, specifically camera placement strategies and movement velocity values, we did not use camera-specific values of *r* or θ. Instead, we obtained averaged values for *r* (5 m) and θ (0.7661 radians) and specified those values for all cameras to intentionally remove effects on density estimates from camera parameter variation. Although this approach negates the use of our resulting density estimates for conservation and management purposes, it was critical for ensuring that our density estimate comparisons between camera placement strategies and movement velocity values were not influenced by variation in the camera parameters (*sensu* Twining et al. [33], who similarly fixed *r* and θ to test the sensitivity of REM density estimates to velocity parameterizations). However, across the entire four-year survey period, some cameras had memory cards that reached capacity, battery life was expended, or the camera was otherwise rendered inoperable for a brief period (e.g., theft, knocked over, etc.); therefore, we did specify the camera-specific operation times in the REM. Additionally, we obtained the mean movement velocity (*v*) among all individuals within a given season × year combination for each species from the CTMM that were fitted to the GPS collar location data, and we specified those values in the REM [47,70]. We fit the REM using bootstrapping implemented via the R package remBoot (v0.1.0) [87], and we produced 95% confidence intervals around point estimates using 1000 iterations.

We conducted the following six REM analyses for the coyote and jackrabbit data within each season × year to evaluate the potential effects on density estimates of using detections from strategically placed cameras and borrowing movement velocity values:Baseline—Used only the detections from randomly placed cameras and the corresponding season × year-specific velocity values. This analysis represents the satisfaction of all REM assumptions.Strategic—Used only the detections from strategically placed cameras and the corresponding season × year-specific velocity values. This analysis represents violation of the REM assumption that animals move independently of camera-traps.Hybrid—Used detections from both the randomly and strategically placed cameras and the corresponding season × year-specific velocity values. This analysis represents partial violation of the REM assumption that animals move independently of camera-traps.Random + Borrowed—Used only the detections from randomly placed cameras but velocity values that were ±20% of the season-specific mean. This analysis represents satisfaction of the REM assumption that animals move independently of camera-traps but conveys the effect of borrowing movement velocity from a different study, time period, or species.Strategic + Borrowed—Used only the detections from strategically placed cameras but velocity values that were ±20% of the season-specific mean. This analysis represents violation of the REM assumption that animals move independently of camera-traps and conveys the effect of borrowing movement velocity from a different study, time period, or species.Hybrid + Borrowed—Used detections from both the randomly and strategically placed cameras and velocity values that were ±20% of the season-specific mean. This analysis represents partial violation of the REM assumption that animals move independently of camera-traps and conveys the effect of borrowing movement velocity from a different study, time period, or species.

We considered the Baseline analysis as the most realistic scenario, or best approximation of ‘truth’, to which we compared density estimates from all other scenarios. Therefore, we calculated the percent relative difference (% RD) from Baseline using the R package SimDesign (v2.12) [88].

### 2.7. Literature Review

We performed a systematic literature review to investigate the prevalence and trends of REM density estimates in published predator–prey ecology studies that were produced using data from the different camera placement strategies and different movement velocity sources. We first conducted independent literature searches via Web of Science (https://www.webofscience.com/) and Google Scholar (https://scholar.google.com) using the following Boolean string: (“random encounter model” OR “random encounter models”). We implemented the final literature searches on 22 January 2024, and we included English and non-English-language articles in our searches. We inspected all returned articles for the following criteria for inclusion in our review: (1) use of the REM to estimate animal density from an empirical camera-trapping dataset; (2) description of camera placement strategy and velocity value source; and (3) publication in a peer-reviewed scientific outlet. We excluded all studies that solely used simulations rather than empirical data, that simply referenced REM but did not apply the method, or that were gray literature (e.g., technical reports, theses, dissertations, etc.). For each REM density estimate, we recorded the year the article was published, the species and taxonomic order for which density was estimated, the International Union for Conservation of Nature and Natural Resources (IUCN) Red List status of the species [89], the camera placement strategy that was employed, the source of the movement velocity value that was used, and whether the study was conducted in a predator–prey ecology framework.

We used generalized linear models to evaluate temporal, camera placement, and velocity value source trends in predator–prey studies that used the REM and predator–prey REM density estimates. First, to quantify the rate of change over time in the number of studies that used the REM (response variable), we fit two models, each with a Poisson or negative-binomial distribution, and included a fixed effect for year (predictor variable; year range: 2008–2023) in both models. Second, to quantify the rate of change over time in the number of REM density estimates (response variable), we fit two models, each with a Poisson or negative-binomial distribution, and included a fixed effect for year (predictor variable; year range: 2008–2023) in both models. Third, to test whether the rate of change over time in the number of REM density estimates differed between camera placement strategy (predictor variable; levels: Random, Strategic, Random + Strategic) and velocity value source (predictor variable; levels: Study, Borrowed), we fit two models for each predictor, each with a Poisson or negative-binomial distribution, and included a fixed-effects interaction between camera placement and year or an interaction between velocity value source and year. For each of the four analyses, we produced estimates from the Poisson or negative-binomial model with the lowest AIC*_c_* value (i.e., the top-ranked, most parsimonious model). We fit the generalized linear models using the R package glmmTMB (v1.1.9) [90]. We based inferences on a combination of model coefficient estimates, predicted marginal effects and their 95% confidence intervals, and *p*-values from post hoc Tukey’s multiple comparisons tests, all of which we obtained using the R packages ggeffects (v1.5.0) and emmeans (v1.10.0) [91,92].

## 3. Results

### 3.1. Animal Capture and Tracking

We captured and outfitted a total of 21 subadult and adult coyotes with GPS collars. An average of 17 coyotes were monitored annually, with 14 individuals being monitored for >1 year. We obtained an average of 29,637 satellite location fixes per data year (November–October) from the collared coyotes (range: 24,499–34,776 locations/data year). The mean number of coyote location fixes obtained per season was 14,166 ± 6735 (mean [$\bar{x}$] ± standard error [SE]) and 15,471 ± 1597 during winter and summer, respectively.

We captured and outfitted a total of 89 jackrabbits with GPS collars; however, collar failures and mortalities (e.g., the collar malfunctioned, and no location data were recorded; capture myopathy occurred <10 days post-capture) limited our final sample size to 67 (75%) subadult and adult jackrabbits. An average of 33 jackrabbits were monitored annually, with no individuals being monitored for >1 data year. After thinning and standardizing the collar monitoring data, we used an average of 10,077 satellite location fixes per data year from the collared jackrabbits (range: 6715–13,438 locations/data year). The mean number of jackrabbit location fixes per season was 5596 ± 1452 ($\bar{x}$ ± SE) and 4481 ± 1910 during winter and summer, respectively.

### 3.2. Movement Velocity Estimation

Model selection supported the IOU or OUF continuous-time movement model as the most parsimonious for 52 seasonal datasets from all 21 individual collared coyotes. Estimated coyote seasonal movement velocities, based on 201–1897 locations per individual within a season × year, ranged from 15.68 to 141.85 km/day, with an average movement velocity across the entire study period of 45.95 km/day ± 3.10 ($\bar{x}$ ± SE; Figure 2). A total of 24 and 28 coyote movement velocity estimates were made for the summer and winter seasons, respectively; mean seasonal coyote movement velocities among data years were 47.89 ± 2.94 and 44.30 ± 5.22 during summer and winter, respectively. Mean season × year coyote movement velocities were 56.23 ± 12.47, 47.74 ± 4.39, 39.52 ± 5.20, and 48.04 ± 4.09 km/day during winter 2019–2020, summer 2020, winter 2020–2021, and summer 2021, respectively.

Model selection supported the IOU or OUF continuous-time movement model as the most parsimonious for 58 seasonal datasets from 67 collared jackrabbits (87%). Estimated jackrabbit seasonal movement velocities, based on 107–996 locations per individual within a season × year, ranged from 1.08 km/day to 97.40 km/day, with an average movement velocity across the entire study period of 8.24 km/day ± 1.81 (Figure 2). A total of 23 and 35 jackrabbit movement velocity estimates were made for the summer and winter seasons, respectively; mean seasonal jackrabbit movement velocities among years were 8.18 ± 4.10 and 8.27 ± 1.41 during summer and winter, respectively. Mean season × year jackrabbit movement velocities were 6.44 ± 0.37, 11.89 ± 7.83, 8.65 ± 1.69, and 4.13 ± 0.83 km/day during winter 2019–2020, summer 2020, winter 2020–2021, and summer 2021, respectively.

### 3.3. Camera-Trapping

Across the entire survey period, we obtained a total of 101 and 1609 unique detections of coyotes at randomly and strategically placed cameras, respectively, and a total of 5249 and 14,191 unique detections of jackrabbits at randomly and strategically placed cameras, respectively (Table 1). Thus, the cameras that were strategically placed to detect coyotes produced 16× and 2.70× more detections of coyotes and jackrabbits, respectively, than did the randomly placed cameras. For coyotes, randomly placed cameras required an average of seven days of operation to produce one detection, whereas strategically placed cameras resulted in an average of two unique detections per day. For jackrabbits, randomly placed cameras produced an average of seven unique detections per day, whereas strategically placed cameras resulted in an average of 20 unique detections per day. Combined across camera placement types, the total seasonal numbers of coyote detections were 500, 424, 468, and 318 during winter 2019–2020, summer 2020, winter 2020–2021, and summer 2021, respectively. Combined across camera placement types, the total seasonal numbers of jackrabbit detections were 9835, 6175, 2471, and 959 during winter 2019–2020, summer 2020, winter 2020–2021, and summer 2021, respectively.

### 3.4. REM Density Estimation

Camera placement—Point estimates of coyote density from the REM that used ‘true’ (CTMM-estimated) velocity values varied substantially depending on camera placement, ranging from 0.04 to 2.10 coyotes/100 km^2^ (Figure 3A). Using the true velocities and detections from either only strategically placed cameras or both randomly and strategically placed cameras caused differences of +1663–2180% and +721–934% RD, respectively, compared to the coyote density estimates based on detections from only randomly placed cameras and true velocities (i.e., Baseline). Point estimates of jackrabbit density from the REM that used true velocity values also varied substantially depending on camera placement, ranging from 0.12 to 1.93 jackrabbits/km^2^ (Figure 3B). Using the true velocities and detections from either only strategically placed cameras or both randomly and strategically placed cameras caused differences of +88–580% and +38–269% RD, respectively, compared to the jackrabbit density estimates based on detections from only randomly placed cameras and true velocities (i.e., Baseline).

Velocity specification—For coyote detections from randomly placed cameras, specifying velocity values that were 20% below or above the mean velocities resulted in REM density estimates that were +25% or −17% RD from Baseline, respectively (Figure 4). For coyote detections from strategically placed cameras, specifying velocity values that were 20% below or above the mean velocities resulted in REM density estimates that were +2119–2745% or +1374–1838% RD from Baseline, respectively. For coyote detections from both randomly and strategically placed cameras combined, specifying velocity values that were 20% below or above the mean velocities resulted in REM density estimates that were +926–1192% or +584–762% RD from Baseline, respectively.

For jackrabbit detections from randomly placed cameras, specifying velocity values that were 20% below or above the mean velocities resulted in REM density estimates that were +25% or −17% RD from Baseline, respectively (Figure 5). For jackrabbit detections from strategically placed cameras, specifying velocity values that were 20% below or above the mean velocities resulted in REM density estimates that were +136–720% or +57–+466% RD from Baseline, respectively. For jackrabbit detections from both randomly and strategically placed cameras combined, specifying velocity values that were 20% below or above the mean velocities resulted in REM density estimates that were +73–361% or +15–207% RD from Baseline, respectively.

### 3.5. Literature Review

A collective 76 journal articles and book chapters via Web of Science and 563 journal articles, book chapters, and gray literature via Google Scholar were obtained from literature searches. A total of 84 journal articles and one book chapter that were published from 2008 through 2023 met inclusion criteria (Appendix A). Those 84 studies produced a total of 789 animal density estimates using the REM, which were for 111 species that represented 11 taxonomic orders (Appendix B).

A total of 14 studies (17%) were conducted in a predator–prey ecology framework, the first of which was published during 2015. Those 14 studies produced 78 total density estimates (10% of all REM density estimates) for 25 species that represented five taxonomic orders. The most REM density estimates in predator–prey ecology studies were produced for wild boar (*n* = 18) and the order Artiodactyla (*n* = 50). A total of 14 REM density estimates (18%) were produced using data from randomly placed camera-traps; however, seven of those estimates (50%) were based on velocity values borrowed from a different study area, species, or time period. A total of nine density estimates (12%) were produced using a velocity value that was obtained for the study species during the study period, but two of those densities were produced using data from strategically placed camera-traps. Thus, a total of seven REM density estimates (9%) from predator–prey ecology studies in the published literature were produced using data that satisfied both the camera placement and movement velocity requirements of the REM. A total of 12 REM density estimates (15%) from predator–prey ecology studies were produced for imperiled species, based on IUCN Red List status, only four of which were produced using data from randomly placed cameras and movement velocities that were estimated for the study species during the study period.

Generalized linear models with Poisson and negative-binomial distributions best fit the number of predator–prey ecology studies that used the REM and the number of REM density estimates in a predator–prey ecology framework, respectively (Appendix A). The number of published predator–prey ecology REM studies increased over time at a rate of λ = 1.31/year (95% CI: 1.11–1.55, *p* = 0. 001; Figure 6A). The number of published REM density estimates in a predator–prey ecology framework increased over time at an average rate of λ = 1.58/year (95% CI: 1.22–1.96, *p* = 0.0005; Figure 6B); that rate differed during 2017–2023 by both camera placement and velocity value source, such that most densities in predator–prey ecology studies were produced using strategically placed cameras (*p* = 0.02–0.04; Figure 6C) and borrowed velocities (*p* = 0.004–0.03; Figure 6D).

## 4. Discussion

The REM was among the first methods developed for estimating density of unmarked wildlife populations, and rapid application of this approach has occurred since the seminal publication in 2008 (Appendix B, Figure A1A,B). However, conflicting results from both empirical and simulation-based studies have caused controversy over the reliability of REM density estimates and the usefulness of those estimates for informing conservation and management [33,93,94,95]. Results from analysis of our multi-year empirical predator–prey dataset demonstrated that using detection data from strategically placed cameras can cause substantial inflation of REM density estimates, and that borrowing movement velocity (day range) values from other studies, time periods, or species can introduce volatility in REM density estimates. Concerningly, our extensive literature review revealed that both of those approaches have been dominant in predator–prey research that applied the REM to estimate densities.

Although some previous studies identified positive bias in REM densities as a consequence of strategic camera placement [38,39], those empirical findings were largely based on either short sampling durations or few camera-traps (e.g., *n* = 9 cameras operated for 63 days; [39]). In contrast, we deployed 83 camera-traps continuously for multiple consecutive years to produce density estimates during ~180-day seasonal periods within each year. A recent review of 34 studies that compared density estimates from the REM with density estimates from other methods (e.g., spatial mark–resight) found an average of ~50% positive bias in REM densities when detection data from strategically placed cameras were used [95]. Our results indicated that the positive bias in REM density estimates from using data obtained at strategically placed cameras could be much more severe, likely depending on the behavior and ecology of the target species and characteristics of the survey design (e.g., number of cameras, survey duration, etc.).

Multiple previous studies used data from camera-traps that were strategically placed to detect a predator, but researchers simultaneously assumed that the same cameras were placed randomly with respect to the prey species (e.g., [42,43]). However, the jackrabbit (prey) detections and density estimates from our empirical study, combined with results from previous studies that compared detection rates of terrestrial mammals between strategically and randomly placed cameras [40,41], indicate that such an assumption is untenable. Despite our strategically placed cameras being located to target coyotes (predator), encounters of jackrabbits at those cameras were more than double the number of encounters at randomly placed cameras, and up to five times greater seasonally, which corresponded to substantial inflation of jackrabbit density estimates. These results highlight a potential problem with the increasingly common approach of repurposing camera-trap data from studies that were designed to detect a particular species and attempting to estimate densities of detected non-target species with the REM (e.g., [49,96]). Within a predator–prey ecology framework, such approaches ignore the reality that, although prey species have habitats available for use that are generally unsuitable for predators (i.e., refugia with low predation risk), some overlap in habitat selection/use and therefore animal movements exist between predators and prey [53]. Consequently, strategic placement of camera-traps to detect a predator or prey species likely will inflate encounters of both species and thus positively bias density estimates from the REM.

Our empirical study results also confirm the findings of Palencia et al. [52] regarding the consequences of borrowing movement velocity values. In a review of studies that compared density estimates from the REM with densities from other methods, a tendency was demonstrated for researchers to use underestimates of movement velocity that, in turn, caused overestimation of density by the REM [95]. In contrast, the recent study by Palencia et al. [52] that investigated the potential effects when non-survey-specific velocity values were used found both positive and negative bias in REM density estimates. Our empirical study results corroborate those findings and also indicate that compounding and severe inflation of REM density estimates can be introduced when data from strategically placed cameras are used in combination with even slightly mis-specified movement velocity values. Study-specific velocities for the target species must be obtained for the survey period to produce minimally biased density estimates with the REM, and methods are available to accomplish this from either the camera-trap data alone or by applying continuous-time movement models that account for tortuosity to GPS collar tracking data, as we did [44,45,46,47]. However, our GPS collar tracking data revealed considerable individual variation in movement velocities for each species, which presumably may be characteristic of many other terrestrial wildlife populations. The formulation of the REM relies on the mean movement velocity during the camera-trapping period (or the product of mean speed and activity level) [44,45], but it is unclear whether the mean is the appropriate metric to mitigate density estimate bias when such large individual variation in velocities exists.

Our predator–prey REM study literature review, although based on a small sample size of studies compared to the broader REM literature, revealed that 91% of REM density estimates that were produced in a predator–prey ecology framework relied on either data from strategically placed cameras, borrowed movement velocity values, or both, which likely severely biased those estimates and raises concerns over the appropriateness of subsequent management actions that may have been implemented in response. Problematically, results from our analysis of those trends across time suggest that the number of predator–prey ecology studies that apply the REM to data that do not meet the requirements may increase in the future. Interestingly, although a similar propensity to borrow movement velocities exists in the broader REM literature (Appendix B, Figure A1D), strategic camera placement appears to be an issue that disproportionately occurs in REM density estimation for predator–prey studies. For instance, among the 711 REM density estimates that were not produced in a predator–prey framework, 23% (*n* = 165) of those densities were based on data from strategically placed cameras, and random camera placement has been the predominant approach applied in the broader REM literature (Appendix B, Figure A1C). The random camera placement requirement of the REM is likely an issue for estimating densities of predators because terrestrial carnivores are often wide-ranging, tend to occupy landscapes at low densities, and do not use landscapes randomly, all of which can result in low detection rates [9,53,97,98]. Across seasons and years, we found that an average of seven consecutive days were required to obtain just one coyote detection at randomly placed camera-traps, whereas strategically placed camera-traps produced an average of two unique coyote detections per day. Thus, if researchers intend to estimate densities of terrestrial carnivores using the REM, our results suggest that multiple consecutive months of camera-trap operation may be needed to obtain sufficient detections while satisfying the random sampling requirement of REM, which could result in violations of other REM assumptions (e.g., population closure [36,37]).

### Limitations

We strongly advise readers against using our empirical coyote and black-tailed jackrabbit REM density estimates for conservation or management purposes, including the estimates from data that satisfied REM requirements (i.e., detections from randomly placed cameras and CTMM-estimated velocity values), for three primary reasons. First, we followed the approach described by Twining et al. [33] that removed variation in camera-specific detection zone parameters and applied the same detection zone parameter values to all cameras. Second, it is unclear whether the GPS collar location fix rates that we used were at short enough intervals to provide accurate movement velocity values in the context of animals moving through the detection zones of cameras. Although multiple previous REM studies used movement velocities that were estimated from GPS collar tracking data with similar fix rates, and our CTMM approach that accounted for tortuosity is considered an acceptable method in the REM framework [95], some evidence suggests that typical GPS collar fix rates may be too coarse, which could result in underestimates of movement velocity [44,47]. As our empirical results demonstrate, the primary consequence of parameterizing the REM with an underestimated velocity value is positive bias in the resulting density estimate. Finally, despite both the random and strategic camera arrays spatially overlapping, potential exists for the strategically placed cameras to have had higher detection rates due to covering a broader spatial extent than the random cameras. However, GPS collar tracking data demonstrated that both coyotes and black-tailed jackrabbits used areas in and near both camera arrays, including locales within the spatially smaller random camera array (Appendix A).

## 5. Conclusions

Rapid development of analytical methods for estimating abundance or density of unmarked wildlife from remote camera-trapping data has occurred over the last two decades, and application of those methods, including in predator–prey ecology studies, is expected to increase in the future [9,32,53]. The utility of those approaches for informing wildlife conservation and management hinges on an understanding of the specific scenarios and sampling conditions in which the methods can produce accurate and reliable parameter estimates [99]. We found that most REM density estimates in previous predator–prey ecology studies are likely unreliable, primarily because movement velocity values were borrowed from other species, studies, or time periods (88%), and/or detection data were used from camera-traps that were strategically placed to detect a species (78%). Analysis of the multi-year empirical predator–prey dataset that we presented clearly conveys the likely consequences on REM density estimates of those widely used approaches, which corroborates findings of other recent research [52]. We implore researchers to not repurpose detection data from strategically placed camera-traps or source velocity values from other studies or species, because doing so likely will cause an unacceptable level of bias in REM density estimates. Importantly, if researchers have detection data from only strategically placed cameras, then the REM should be removed from consideration for density estimation, and other analytical approaches that do not necessitate random camera placement should be used [32,33,94].

## Figures and Tables

**Figure 1 animals-14-03361-f001:**
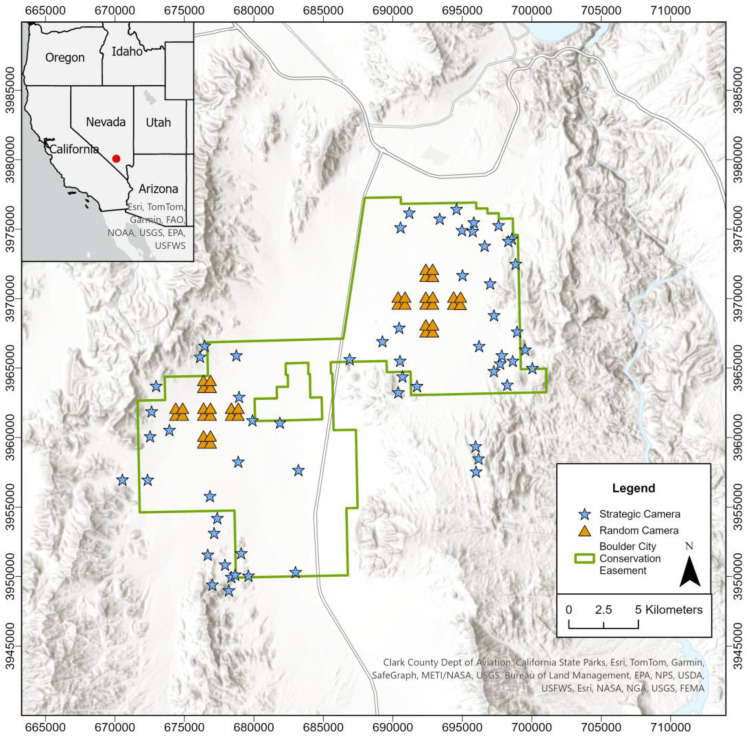
The study area was centered on the Boulder City Conservation Easement in southern Nevada, United States, located within the Mojave Desert. Randomly and strategically placed remote camera-traps were deployed to obtain detections of coyotes (*Canis latrans*; predator) and black-tailed jackrabbits (*Lepus californicus*; prey) for estimating their population densities using the random encounter model (2019–2021). Universal Transverse Mercator (UTM) coordinates (meters; Zone 11) are presented around the border.

**Figure 2 animals-14-03361-f002:**
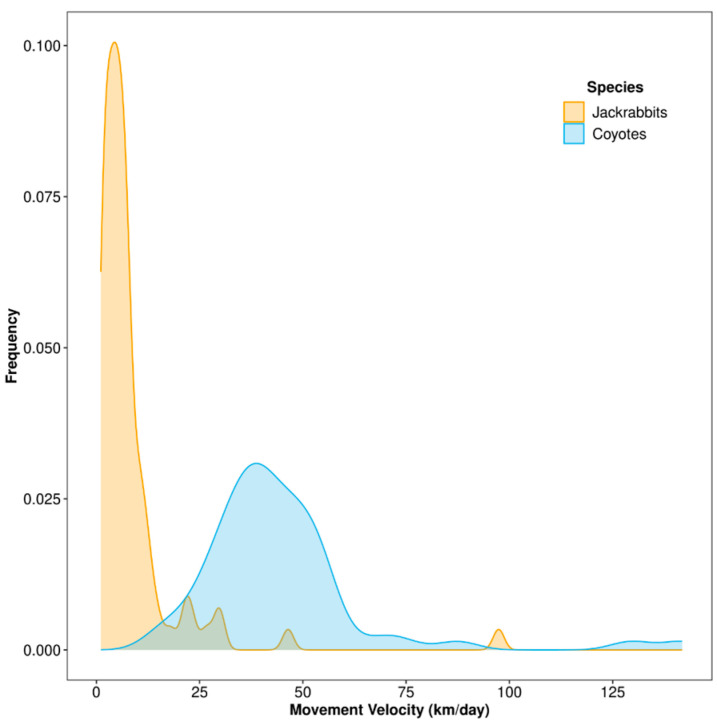
Distributions of movement velocities (day ranges) estimated by continuous-time movement models using locations from collared coyotes (*Canis latrans*; *n* = 52 season × year estimates) and black-tailed jackrabbits (*Lepus californicus*; *n* = 58 season × year estimates) within the Mojave Desert in southern Nevada, United States (2019–2021).

**Figure 3 animals-14-03361-f003:**
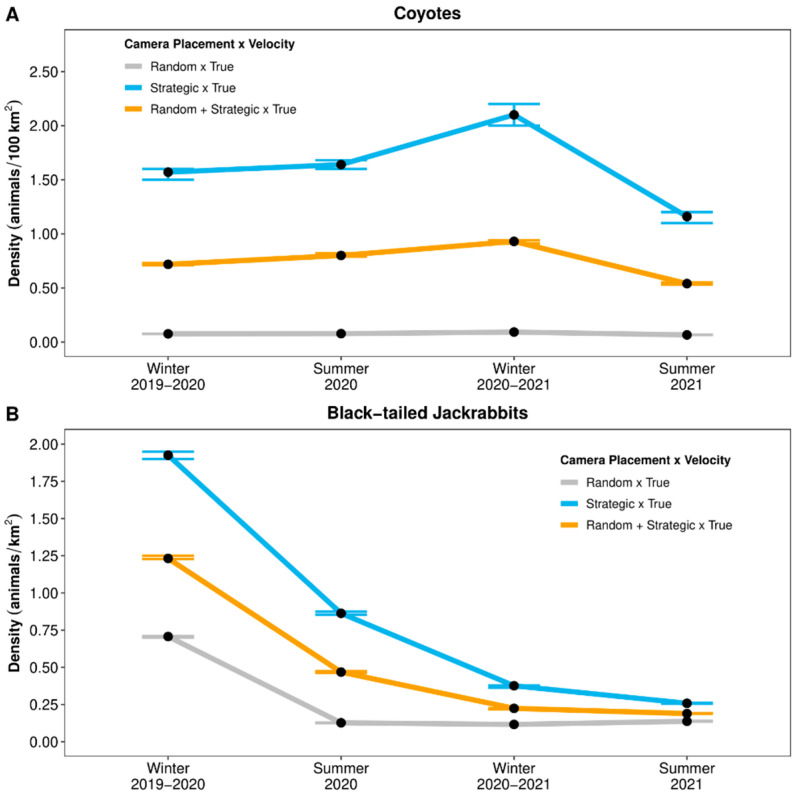
Point estimates (black dots) and 95% confidence intervals (error bars) of seasonal densities for (**A**) coyotes (*Canis latrans*; per 100 km^2^) and (**B**) black-tailed jackrabbits (*Lepus californicus*; per km^2^) within the Mojave Desert in southern Nevada, United States. Densities were estimated via random encounter models that used species × season × year-specific (i.e., ‘true’) movement velocities (day ranges), which were estimated from continuous-time movement models, and used detection data from randomly, strategically, and randomly + strategically placed camera-traps (2019–2021).

**Figure 4 animals-14-03361-f004:**
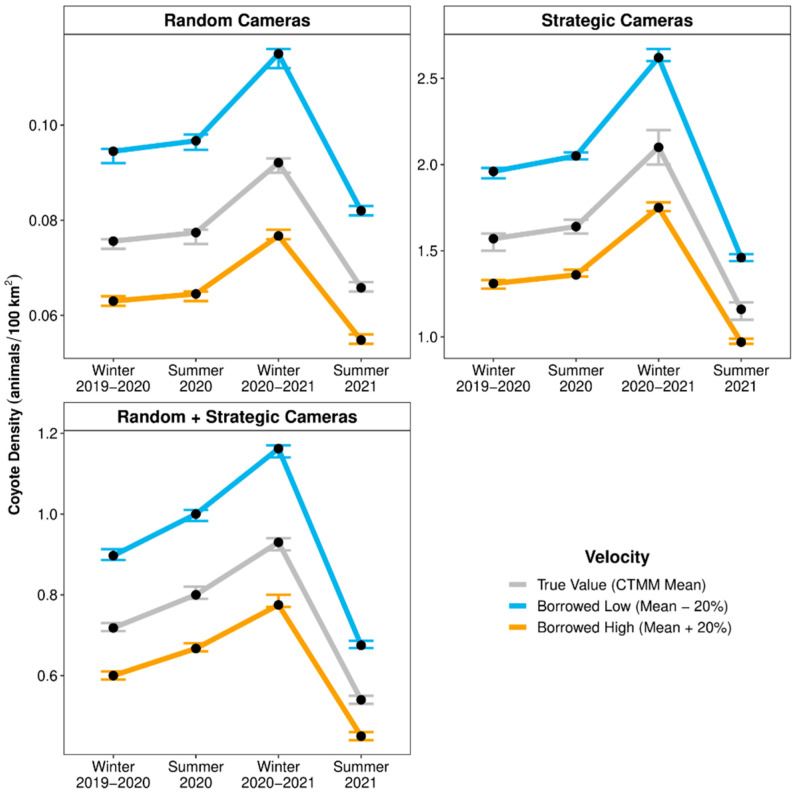
Point estimates (black dots) and 95% confidence intervals (error bars) of seasonal densities for coyotes (*Canis latrans*; per 100 km^2^) within the Mojave Desert in southern Nevada, United States, estimated by random encounter models that used movement velocities (day ranges) that were ±20% from the season-specific mean velocities estimated by continuous-time movement models (CTMM) and used detection data from randomly, strategically, and randomly + strategically placed camera-traps (2019–2021).

**Figure 5 animals-14-03361-f005:**
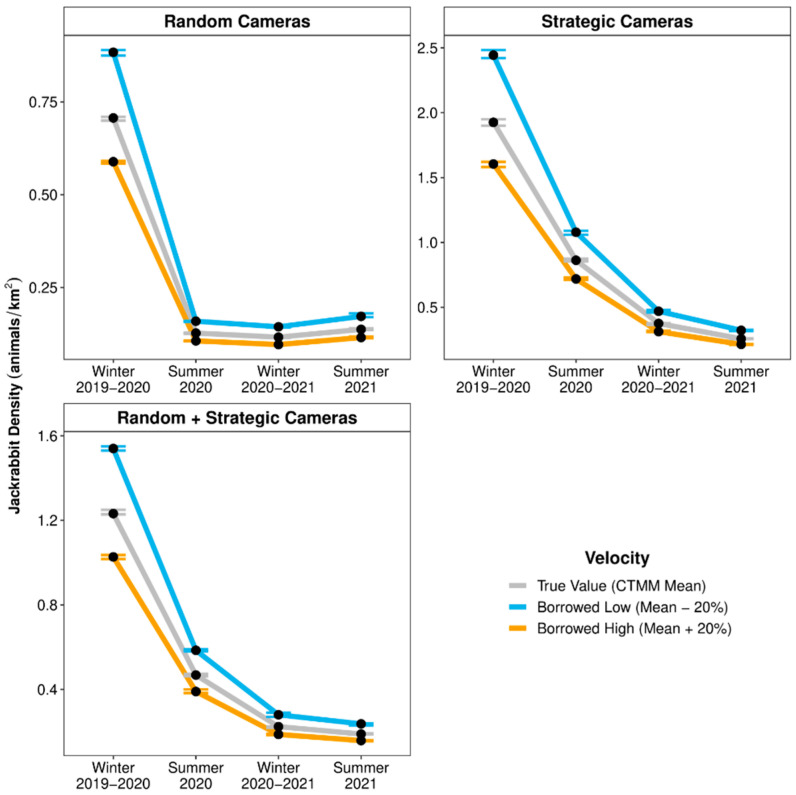
Point estimates (black dots) and 95% confidence intervals (error bars) of seasonal densities for black-tailed jackrabbits (*Lepus californicus*; per km^2^) within the Mojave Desert in southern Nevada, United States, estimated by random encounter models that used movement velocities (day ranges) that were ±20% from the season-specific mean values estimated by continuous-time movement models (CTMM) and used detection data from randomly, strategically, and randomly + strategically placed camera-traps (2019–2021).

**Figure 6 animals-14-03361-f006:**
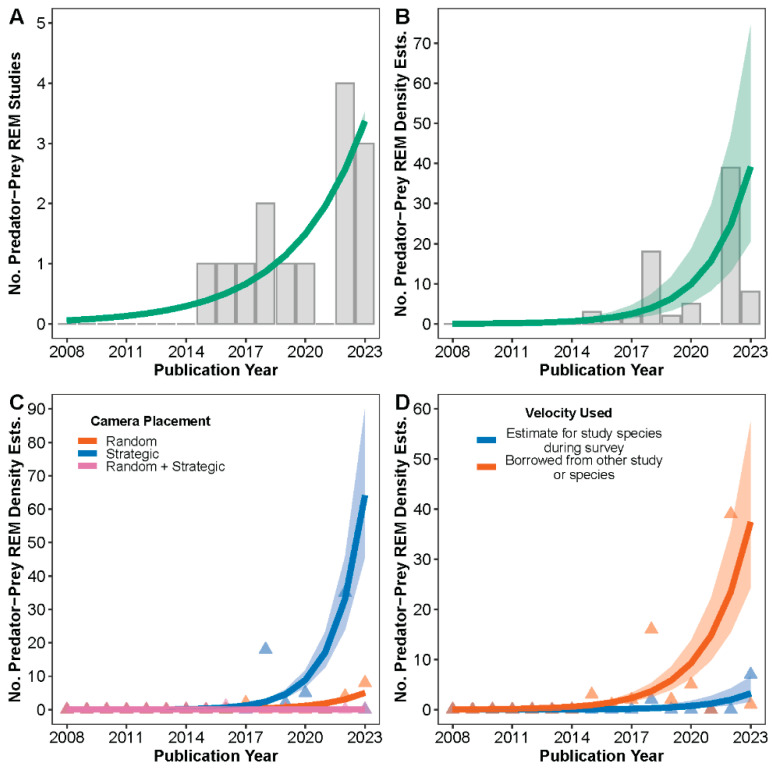
Predicted marginal effects point estimates (solid lines) and 95% confidence intervals (shaded regions) for (**A**) the number of published predator–prey ecology studies that used the random encounter model (REM), (**B**) the number of REM density estimates produced by predator–prey ecology studies, (**C**) the number of REM density estimates from predator–prey ecology studies by camera placement strategy, and (**D**) the number of REM density estimates from predator–prey ecology studies by movement velocity (day range) value used. Results are from generalized linear models with negative-binomial or Poisson error distributions. Raw counts are presented as background columns or triangles.

**Table 1 animals-14-03361-t001:** Summary of the number of unique detections of coyotes (*Canis latrans*) and black-tailed jackrabbits (*Lepus californicus*) at randomly and strategically placed cameras, which were used in random encounter models to estimate seasonal densities during 2019–2021 within the Mojave Desert in southern Nevada, United States.

	Coyote Detections		Jackrabbit Detections	
Season × Year	Random	Strategic	Random	Strategic
Winter 2019–2020	30	470	3214	6621
Summer 2020	22	402	898	5277
Winter 2020–2021	27	441	742	1729
Summer 2021	22	296	395	564

## Data Availability

The camera-trap detection data are openly available in the Dryad Digital Repository at: https://doi.org/10.5061/dryad.djh9w0w6v.

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
