# Peer review of "Most Random-Encounter-Model Density Estimates in Camera-Based Predator–Prey Studies Are Unreliable"

_animals, 2024, doi:10.3390/ani14233361_

Round 1
Reviewer 1 Report
Comments and Suggestions for Authors
The authors present REM density estimates for coyote and jackrabbit based on camera surveys using strategically placed, and randomly placed camera locations. Capitalizing on simultaneous GPS tracking of both species, the authors compare density estimates using high quality data that meets the assumptions of REM models, with estimates based on potentially biased data, using the strategically placed cameras, purposely biased movement estimates, or some combination of them. The overall goal is to show the impact on density estimates of not meeting full assumptions of the REM model. The authors also show the relatively frequent publication of density estimates using methods/data that have not met REM assumptions, based on an extensive literature review. Although other papers have investigated the potential biases involved in relaxing various assumptions of the REM models, which for various reasons are challenging to meet, the authors present here an important additional data point in this overall discussion, particularly with the relatively novel combination of having both tracking data for the study species AND having both a random and species targeted camera layout. The analyses presented, for the most part, take advantage of this data set and the resulting conclusions are well supported.
Although detailed below, my most critical suggestions involve the lack of detail and potential lack of robustness of the GLMM analyses focused on temporal trends in REM analyses and I make some suggestions for focusing that.
I am also concerned about enormous individual variation in movement velocities estimated from the tracking data and request additional discussion of the role this could play in interpreting results from REM models given the extent of this variation. Relatedly, it is not clear why, as their point of comparison, the authors did not at least present the range of movement velocities already published for these two species, given that use of these numbers would serve as a more direct representation of the common approach of “borrowing” velocity data from the literature. The selection of the 20% higher and lower values for velocities appears arbitrary, particularly given that there appears to be much larger variation across individuals within this one study.
Finally the lack of spatial overlap between the random grid and the strategic cameras requires at least some discussion in the paper as I discuss below. This layout seems to run the risk that the ACTUAL density of the two species is functionally different between the two camera sampling designs, which would bias the results of nearly every aspect of the presented analyses.
INTRODUCTION
I think given the scope of the paper and the inclusion of a wider review of applications of REM in the literature, it makes sense to at least mention the competing methods of density estimation of unmarked individuals that exist. The relative complexity of some, for example the need to estimate distance from the camera for the distance sampling approach, will likely support your statement that REM use is likely to continue if not increase. But right now the reader would not realize there are multiple other approaches (e.g. time to detection) available from what is presented.
Line 172: “predating” seems more concise and likely more accurate than “depredating on” here.
METHODS
Why is there no overlap between the random cameras and the strategically placed cameras? I would have assumed the two sampling designs would be sampling the same area. That is broadly true, but the strategically placed cameras seem to be sampling potentially different habitat, closer to the more complex topography of the study area. It must at least be discussed in the paper what the potential consequences are of this layout, particularly if movement data indicated coyotes largely did not use the areas where the grids were established (this would be a critical flaw that would be hard to overcome). Overlaying tracking data on the study figure, or providing this as a supplementary figure, might be helpful.
Line 276: What behavioral or biological evidence can you present to indicate that the sampling interval of the coyotes and jackrabbits is not too course? I understand that the ctmm approach allows one to quantitatively assess whether enough information exists to estimate movement speeds, but 3 hours seems like a very course resolution to sample movement of a coyote, regardless of what the ctmm movement model assesses. Did the authors consider estimating movement velocity only from the higher resolution data, or at least comparing results from the two sampling intervals? I recognize ctmm models should be able to deal appropriately with different sampling intervals, and you note that the models account for impacts of tortuosity. But the tortuosity comes into play primarily with ultra high resolution data, which the authors do not have, and the models cannot recreate tortuosity when sampling is too course.
Line 401: It’s not totally clear what the goal of the GLMMs was here, or what the response variables and potential predictor variables were. You mention interactions with year, but it seems year may be the only predictor variable? If there are others it is not clear what they are. Make clear what are the response and predictor variables used, and what model combinations are tested if there is more than one predictor. If year is the only predictor it is not clear what post-hoc multiple comparisons would be used for. Unless this is referring to another statistical test entirely. Also how was it decided which distribution to use? Presumably a test of overdispersion but the threshold and the test used should be specified.
It seems to me that the # of density estimates generated in a paper is much less relevant than the number of papers published. A given paper seems very likely to have used the same analytical approach across all density estimates. This results in each density estimate inside a paper not being an independent sample/decision about what analytical approach to use, and therefore much less robust as a sampling/survey approach. I would strongly suggest removing the analysis of each density estimate as a result. In addition, 14 is a very low sample size for the predator prey data set to investigate for temporal trends. I cannot think of a reason why this particular type of study would employ different REM methods on average than the REM literature in general. And the larger dataset is a much more defensible approach to looking at temporal trends. So I suggest only looking at trends over time in REM usage, using the full dataset (move Appendix results in some form into the full paper). In addition, from there the interesting question seems to be what PROPORTION of the published REM papers are using appropriate methods/data. If there are more papers published using REM, then the # using borrowed velocities or strategically placed cameras will likely also increase. It seems more interesting to ask whether researcher are becoming smarter about meeting REM assumptions or not. If you can show they are not, then you can assume predator prey studies will continue to suffer from poor estimates. Finally I also noticed that the time block from 2017-2023 was somehow separated in some analyses. That should be detailed in the methods as well. It’s not clear if this was done by running a totally separate model, or having this as a categorical predictor with an interaction term for example.
RESULTS
Line 413: Here you present average total number of locations collected within season/year combinations. This implies that the analysis of the movement data will pool locations across individuals. However in the paragraphs below it seems that movement velocities were estimated at an individual level and then averaged. If so, this should be clarified in the methods, and in this part of the results it would be more relevant to report the range of locations at an individual level, unless there are also some analyses that pool locations across individuals.
Line 422: What was a “sufficient” amount of monitoring data to include an individual in the analysis?
Figure 2. I think the figure or the caption could benefit from sample size information for the two species. I assume the relevant sample size here would be season x year combinations across individuals?
Line 444: This is perhaps for the discussion section, but how is one to interpret this enormous range of movement velocities within the species? One could argue that an average of 8 km/day is misleading and potentially not very useful given such an enormous range of values…if the authors have an idea as to why this range is so large, it would be worth including in the discussion, particularly since the range is greater than the 20% variation you introduce to simulate pulling data from other studies for this value. Would there be good reason to use a median value when the range is so large? Is there a case to be made for removing outliers here?
Line 459: I’m not sure a statistical test is necessarily required or useful here, but the test should be described in the methods and I don’t recall seeing that.
DISCUSSION
The supplementary materials statement on line 666 does not make mention of the appendix. Although this journal seems to have a free formatting guideline, it does not mention Appendices, so I suggest moving the Appendix to supplemental materials. Presumably the editor will have a suggestion here in either case.
REFERENCES
Reference 32 is missing a date
Author Response
Reviewer 1
The authors present REM density estimates for coyote and jackrabbit based on camera surveys using strategically placed, and randomly placed camera locations. Capitalizing on simultaneous GPS tracking of both species, the authors compare density estimates using high quality data that meets the assumptions of REM models, with estimates based on potentially biased data, using the strategically placed cameras, purposely biased movement estimates, or some combination of them. The overall goal is to show the impact on density estimates of not meeting full assumptions of the REM model. The authors also show the relatively frequent publication of density estimates using methods/data that have not met REM assumptions, based on an extensive literature review. Although other papers have investigated the potential biases involved in relaxing various assumptions of the REM models, which for various reasons are challenging to meet, the authors present here an important additional data point in this overall discussion, particularly with the relatively novel combination of having both tracking data for the study species AND having both a random and species targeted camera layout. The analyses presented, for the most part, take advantage of this data set and the resulting conclusions are well supported.
Response: Thank you for providing a thorough, helpful review of our manuscript.
Although detailed below, my most critical suggestions involve the lack of detail and potential lack of robustness of the GLMM analyses focused on temporal trends in REM analyses and I make some suggestions for focusing that.
Response: We added details to the Methods subsection for the analysis of the REM lit review data with GLMMs.
I am also concerned about enormous individual variation in movement velocities estimated from the tracking data and request additional discussion of the role this could play in interpreting results from REM models given the extent of this variation. Relatedly, it is not clear why, as their point of comparison, the authors did not at least present the range of movement velocities already published for these two species, given that use of these numbers would serve as a more direct representation of the common approach of “borrowing” velocity data from the literature. The selection of the 20% higher and lower values for velocities appears arbitrary, particularly given that there appears to be much larger variation across individuals within this one study.
Response: First, we were also somewhat shocked at the large amount of individual variation in movement velocities. However, we perceive this not as a negative but as an opportunity to convey that if such variation is present in other terrestrial wildlife populations, which the formulation of the REM is incapable of reflecting, then reliability of resulting REM density estimates should be further scrutinized. Second, we searched the literature for movement velocities estimated for both coyotes and black-tailed jackrabbits and found few values produced for coyotes (n = 11; range = 3.43–40.80 km/day), most of which were derived and not estimated via a formal modeling framework, and no velocities have been produced for black-tailed jackrabbits. All those reported values for coyotes were lower than all our season × year-specific velocity estimates, thereby precluding their utility to our analysis because no velocities greater than ours were available. Thus, considering there were no velocities in the literature for black-tailed jackrabbits and all coyote velocities in the literature were less than ours, we chose to use the 20% lower and higher adjustments to our model-estimated velocities. Although we agree that the 20% threshold is somewhat arbitrary, previous studies have used as much as 50% adjustments (Jensen et al. 2022; https://doi.org/10.1093/jmammal/gyac009), which we perceived as being too large and possibly not reflective of most studies that used borrowed velocity values.
Finally the lack of spatial overlap between the random grid and the strategic cameras requires at least some discussion in the paper as I discuss below. This layout seems to run the risk that the ACTUAL density of the two species is functionally different between the two camera sampling designs, which would bias the results of nearly every aspect of the presented analyses.
Response: We disagree that spatial overlap did not exist between the two camera placement types; however, we agree that the strategic cameras surveyed a broader spatial extent than the random cameras. We added a discussion of this in a new Limitations subsection of the revised manuscript and we also included a supplemental figure (Figure S1) that shows the GPS collar locations for both coyotes and jackrabbits relative to the locations of random and strategic cameras.
INTRODUCTION
I think given the scope of the paper and the inclusion of a wider review of applications of REM in the literature, it makes sense to at least mention the competing methods of density estimation of unmarked individuals that exist. The relative complexity of some, for example the need to estimate distance from the camera for the distance sampling approach, will likely support your statement that REM use is likely to continue if not increase. But right now the reader would not realize there are multiple other approaches (e.g. time to detection) available from what is presented.
Response: We added mention of other available methods for density estimation of unmarked populations while referring readers to previous studies that covered comparisons among those approaches for further details.
Line 172: “predating” seems more concise and likely more accurate than “depredating on” here.
Response: We have retained “depredating” here and refer the reviewer to the Ecological Society of America’s publication entitled, “To predate or depredate: What’s the word?” (https://www.jstor.org/stable/bullecosociamer.87.2.128). We note that ‘depredate’ has historically been the scientific term used to describe “to prey upon” and has been used for that purpose since 1651. In contrast, ‘predate’ describes “to seek prey” and first appeared in 1974.
METHODS
Why is there no overlap between the random cameras and the strategically placed cameras? I would have assumed the two sampling designs would be sampling the same area. That is broadly true, but the strategically placed cameras seem to be sampling potentially different habitat, closer to the more complex topography of the study area. It must at least be discussed in the paper what the potential consequences are of this layout, particularly if movement data indicated coyotes largely did not use the areas where the grids were established (this would be a critical flaw that would be hard to overcome). Overlaying tracking data on the study figure, or providing this as a supplementary figure, might be helpful.
Response: We understand the reviewer’s concern, and we have added discussion about the spatial extent of the two camera placement types, including potential issues, to the Limitations subsection of the revised manuscript. Additionally, we have provided a supplementary figure (Figure S1) that shows all GPS collar locations for both species during our study period relative to the camera traps.
Line 276: What behavioral or biological evidence can you present to indicate that the sampling interval of the coyotes and jackrabbits is not too course? I understand that the ctmm approach allows one to quantitatively assess whether enough information exists to estimate movement speeds, but 3 hours seems like a very course resolution to sample movement of a coyote, regardless of what the ctmm movement model assesses. Did the authors consider estimating movement velocity only from the higher resolution data, or at least comparing results from the two sampling intervals? I recognize ctmm models should be able to deal appropriately with different sampling intervals, and you note that the models account for impacts of tortuosity. But the tortuosity comes into play primarily with ultra high resolution data, which the authors do not have, and the models cannot recreate tortuosity when sampling is too course.
Response: Few movement velocity values have ever been produced for coyotes (n = 11, most of which were derived, not estimated) and none have ever been produced for black-tailed jackrabbits. Consequently, insufficient evidence exists to either confirm or refute that the GPS-collar fix intervals we used were inadequate for estimating movement velocities of either species. Therefore, following the approach of Noonan et al. (2019) and only including velocity values that were estimated from an IOU or OUF continuous-time movement model when model selection supported one of those is the best tool we had available. We did not attempt to estimate coyote velocities from the hourly fix data that were obtained during the first five days of each month because that five-day duration did not represent the survey period nor were coyote detections at camera-traps sufficient within a five-day period to produce REM density estimates. Further, we noted in the original draft of the manuscript that the resulting REM density estimates should not be used for conservation or management, and we have expanded on this by adding a Limitations paragraph to the Discussion section.
Line 401: It’s not totally clear what the goal of the GLMMs was here, or what the response variables and potential predictor variables were. You mention interactions with year, but it seems year may be the only predictor variable? If there are others it is not clear what they are. Make clear what are the response and predictor variables used, and what model combinations are tested if there is more than one predictor. If year is the only predictor it is not clear what post-hoc multiple comparisons would be used for. Unless this is referring to another statistical test entirely. Also how was it decided which distribution to use? Presumably a test of overdispersion but the threshold and the test used should be specified.
Response: We added details to this subsection in the Methods to clarify what the response and predictor variables were. Specifically, either the number of studies or number of density estimates were the response variable; an interaction between year and either camera placement or velocity source were the predictor variables. Model selection based on AICc was used to determine which model and therefore distribution was most supported for each response and predictor variables combinations (i.e., a Poisson model and a negative-binomial model was fit for each response and predictor combination, and AICc was used to determine which model to produce parameter estimates from).
It seems to me that the # of density estimates generated in a paper is much less relevant than the number of papers published. A given paper seems very likely to have used the same analytical approach across all density estimates. This results in each density estimate inside a paper not being an independent sample/decision about what analytical approach to use, and therefore much less robust as a sampling/survey approach. I would strongly suggest removing the analysis of each density estimate as a result. In addition, 14 is a very low sample size for the predator prey data set to investigate for temporal trends. I cannot think of a reason why this particular type of study would employ different REM methods on average than the REM literature in general. And the larger dataset is a much more defensible approach to looking at temporal trends. So I suggest only looking at trends over time in REM usage, using the full dataset (move Appendix results in some form into the full paper). In addition, from there the interesting question seems to be what PROPORTION of the published REM papers are using appropriate methods/data. If there are more papers published using REM, then the # using borrowed velocities or strategically placed cameras will likely also increase. It seems more interesting to ask whether researcher are becoming smarter about meeting REM assumptions or not. If you can show they are not, then you can assume predator prey studies will continue to suffer from poor estimates. Finally I also noticed that the time block from 2017-2023 was somehow separated in some analyses. That should be detailed in the methods as well. It’s not clear if this was done by running a totally separate model, or having this as a categorical predictor with an interaction term for example.
Response: We disagree, primarily because multiple studies produced multiple density estimates for multiple species using both random and strategic camera placements and/or borrowed and study-specific velocity values, which varied by species and study area within a given study. This can be seen in the supplementary Spreadsheet S1 of the literature review results that we provided. For example, Caravaggi et al. (2016) produced 5 density estimates for two species, one species of which they obtained velocities for using data from the study area, the other species of which they borrowed velocities for from a separate study in a different area. Similarly, Rahman et al. (2017) produced 6 density estimates for one species, but 2 estimates were from strategically placed cameras, 2 estimates were from randomly placed cameras, and 2 estimates were from both strategically and randomly placed cameras. Thus, by focusing on the study alone, instead of also the density estimates, such discrepancies would be lost and paint an inaccurate picture of what has occurred. Further, as is evident by comparing the predator-prey REM study results (Figure 6) and the broader REM study results (Figure A1), the trends in inappropriate usage of camera placement types are different, whereas the trends in velocity value source are similar. Thus, the reviewer’s assumptions about predator-prey REM studies relative to the broader REM studies are unsupported and we have retained the analysis and results as presented in the original manuscript. Finally, there was no time block during 2017-2023 that was separated in some analyses and that is not conveyed on any of the figures. We suspect the reviewer is referring to what appears to be missing data in the background columns for the year 2021 on Figure 6A and 6B, but that is simply the accurate reflection that no REM density estimates were produced in the predator-prey framework during 2021, which we correctly recorded in our literature review dataset as a 0 value for year 2021.
RESULTS
Line 413: Here you present average total number of locations collected within season/year combinations. This implies that the analysis of the movement data will pool locations across individuals. However in the paragraphs below it seems that movement velocities were estimated at an individual level and then averaged. If so, this should be clarified in the methods, and in this part of the results it would be more relevant to report the range of locations at an individual level, unless there are also some analyses that pool locations across individuals.
Response: Movement velocities were estimated at the individual-level and then averaged within each season × year combination, which we reported in the Methods section of the original manuscript (lines 349-352). Given the REM requires mean velocity values within the timeframe of camera sampling (e.g., season), we believe that it is more relevant to report the number of locations for each season.
Line 422: What was a “sufficient” amount of monitoring data to include an individual in the analysis?
Response: We further clarified that collar failures (i.e., the collar malfunctioned and no location data were recorded, the battery died and the collar was never recovered) and mortalities (i.e., capture myopathy within 10 days of capture) prevented velocity estimations because of small sample sizes for locations.
Figure 2. I think the figure or the caption could benefit from sample size information for the two species. I assume the relevant sample size here would be season x year combinations across individuals?
Response: We provided sample sizes for both species in the revised figure caption.
Line 444: This is perhaps for the discussion section, but how is one to interpret this enormous range of movement velocities within the species? One could argue that an average of 8 km/day is misleading and potentially not very useful given such an enormous range of values…if the authors have an idea as to why this range is so large, it would be worth including in the discussion, particularly since the range is greater than the 20% variation you introduce to simulate pulling data from other studies for this value. Would there be good reason to use a median value when the range is so large? Is there a case to be made for removing outliers here?
Response: We interpret the movement velocity estimates as likely being the reality for most terrestrial mammal populations, such that some individuals have established home ranges and exhibit slower movement velocities (e.g., residents), whereas other individuals do not have established home ranges and exhibit faster movement velocities (e.g., transients/dispersers). In light of those space use characteristics in terrestrial mammal populations, we agree that using the mean value in the REM is likely misleading, particularly because the range of variation in observed velocities is disregarded. However, the REM requires a mean velocity value across the period of camera-trap operation, so that is what we used. We have mentioned this in the revised Discussion.
Line 459: I’m not sure a statistical test is necessarily required or useful here, but the test should be described in the methods and I don’t recall seeing that.
Response: We removed the chi-square tests from the Results section.
DISCUSSION
The supplementary materials statement on line 666 does not make mention of the appendix. Although this journal seems to have a free formatting guideline, it does not mention Appendices, so I suggest moving the Appendix to supplemental materials. Presumably the editor will have a suggestion here in either case.
Response: We followed the journal’s specified guidelines for Appendices and Supplemental Information, and the Appendix was appropriately referenced in the main text of the manuscript.
REFERENCES
Reference 32 is missing a date
Response: Thank you. We corrected this.
Reviewer 2 Report
Comments and Suggestions for Authors
This study provides long-term empirical camera trapping data to show that strategically deployment of camera traps or using velocity borrowed from other studies cannot provide reliable estimation of population density using REM method. These two violations were emphasized all the time in the past. The criticism of previous publications is good and important. Several questions or comments relate to the accuracy of the density estimation are provided here, although accuracy may not be the most important issue for this paper:
1. The distance between camera trappers of the random set is very close (only 500m), which might violate the independency requirement considering the large home range (>25km2) of coyotes. Also, please provide the distance between camera trappers for the strategic set.
2. The paper doesn’t mention the activity level adjustment in the method section. No need?
3. The use of the “averaged values for r (5 meters) and θ (0.7661 radians)” for all camera sites eliminated the variation amount sites will introduce bias when overall density and local population size are desirable.
Author Response
This study provides long-term empirical camera trapping data to show that strategically deployment of camera traps or using velocity borrowed from other studies cannot provide reliable estimation of population density using REM method. These two violations were emphasized all the time in the past. The criticism of previous publications is good and important. Several questions or comments relate to the accuracy of the density estimation are provided here, although accuracy may not be the most important issue for this paper:
- The distance between camera trappers of the random set is very close (only 500m), which might violate the independency requirement considering the large home range (>25km2) of coyotes. Also, please provide the distance between camera trappers for the strategic set.
Response: There is no specified minimum distance between camera-traps for use in the REM, nor is independence among camera-traps an assumption of the REM. Instead, each detection at a given camera-trap is assumed to be an independent event.
- The paper doesn’t mention the activity level adjustment in the method section. No need?
Response: An activity level adjustment is not required for fitting the REM to estimate density. However, activity level may or may not be required only for estimating the movement velocity (day range) parameter, depending on which velocity estimation method is used. Our approach using continuous-time movement models applied to GPS-collar location data does not require a separate activity level adjustment to estimate velocities.
- The use of the “averaged values for r (5 meters) and θ (0.7661 radians)” for all camera sites eliminated the variation amount sites will introduce bias when overall density and local population size are desirable.
Response: Correct, which is why we stated in the original manuscript that our resulting density estimates for both species should not be used for conservation or management purposes, but that this approach of removing detection zone variation among cameras was necessary to enable direct comparisons between camera placement strategies and velocity values. This is a similar approach to what Twining et al. (2022) used.
Reviewer 3 Report
Comments and Suggestions for Authors
A really nice manuscript that dives into Random Encounter Modelling (REM), a relatively new density estimation method, and investigates its weaknesses and potential widespread misuse. Overall, I really like the combination of empirical data analysis with the literature review approach, and congratulate the authors on an interesting study that uses diverse methods to demonstrate the limitations of the REM approach, as it increases in popularity. The movement data set is extensive, and the camera data though spatially limited was deployed over a long time period, and appears sufficient to assess the manuscript’s research questions. I have some minor comments on the intro and discussion that might help with manuscript flow. Some more information on the GLM modelling component would increases transparency and reproducibility of the methods. The literature review component could be pushed slightly further for more impact. While information on all REM modelling papers is collected, the results and discussion focus only on the presentation of the REM papers that consider predator-prey systems. It’d be great to see an assessment of the REM assumption violation across all REM modelling literature- rather than just the focus on the small subset. Overall though a great read, thanks for the opportunity to review.
Line 18: Arguably all species do not use landscapes randomly- perhaps delete ‘predator’ here.
Line 30: A bit of detail lacking here for the uninitiated. Perhaps define the issue more clearly by changing to: ‘… lack individual unique natural markings required to undertake standard density estimation analyses’.
Line 63: ref?
Line 77: The terminology is a little tricky here. Mark resight methods are used in camera trapping studies where the animals are individually identifiable, in that individuals ‘marked’ when their patterns are first seen on camera and ‘resighted’ when the patterns are reseen on subsequent images. Also, physical capture/marking methods were developed prior to the widespread use of camera trapping. Perhaps change this sentence to: Approaches have been used to resolve this individual…’ and line 84 to: ‘However such approaches…’.
Line 85: I’m not sure this paragraph on indices is needed. Indices have their place, but they don’t generally provide estimates for density which is what is being tested/discussed here ... To tighten the intro and keep it flowing, I’d suggest deleting this para apart from the last sentence, and using that as the first sentence in the next paragraph.
Line 105: I’d link these next two paragraphs together and whittle it down a bit if you can, as they really talk about the same thing. That’d produce a tighter 5 paragraph intro.
Line 130: I think you could finish in the final part of this para with a statement about the unknowns that your study will answer. Currently the lit review ideas mentioned at line 141 are a bit out of the blue, here you could set the reader up to expect this. For example, add a sentence that states something along the lines of: ‘To date there have been few empirical tests of movement velocity and camera placement impacts on REM density estimates, and no assessment of the extent this is an issue in REM model implementation in the predator-prey literature’.
Line 202: Really nice methods.
Line 389: State if non-english language papers were considered.
Line 402: I’m a little confused here about the model structure. What are the response and predictor variables in each model? Have you run three models each with a different response variable and the same predictors? Also describe the model selection process- have you just run the global model or run a model set? Perhaps a table in the supplementary material describing the models, variables included and the AIC values (ect) would help.
Line 534: I could be missing something here but to me it looks like the search was conducted for REM papers generally, but the results are focused on the presentation of REM studies that are focused on a predator-prey system. While this is relevant to the current study, it would be great to also have info on the broader REM literature and how misleading that is. This seems particularly important given the sample sizes involved (84 REM studies, but only 17 REM studies in a predator-prey framework) and with the literature collated there is a good opportunity to do this.
Line 551: Again, slightly more info on modeling approach is needed- what was the model structure, and if there were competing models in the same model set, how much better was the best fitting model? Perhaps add a table even in the supplementary material to lay it out.
Line 572: There’s an opportunity with the first paragraph of the discussion to grab the readers attention and to tell them the main findings. This has been done well in the concluding paragraph, but would be useful to do here. As it stands the reader needs to wade through this paragraph before getting to any key findings or results.
Line 593: This is a bit of a tangent, but I wonder for species like coyotes that use habitat in a very nonrandom way (choosing to move along features like tracks and roads) - how accurate are REM estimates likely to be even if random detectors arrangements are used? Eg are there species that this method isn’t appropriate for just because of their biology? I feel like the missing piece here is to compare density estimates between SECR and REM for species that use the landscape in a more or a less random way. This is just more of a thought bubble no need to change anything..
Line 606: It seems like there were also season differences in the extent this occurred, perhaps worth mentioning- up to you.
Line 635: Here there is a discussion of the finding that most REM density estimates in predator prey literature are misleading- that’s super interesting. It would be great to also see a comparison and discussion of these figures generally (outside of predator and prey literature). Do the density estimates in the broader REM literature also rely on erroneous estimates of velocity and strategic camera placements?
Author Response
Reviewer 3
A really nice manuscript that dives into Random Encounter Modelling (REM), a relatively new density estimation method, and investigates its weaknesses and potential widespread misuse. Overall, I really like the combination of empirical data analysis with the literature review approach, and congratulate the authors on an interesting study that uses diverse methods to demonstrate the limitations of the REM approach, as it increases in popularity. The movement data set is extensive, and the camera data though spatially limited was deployed over a long time period, and appears sufficient to assess the manuscript’s research questions. I have some minor comments on the intro and discussion that might help with manuscript flow. Some more information on the GLM modelling component would increases transparency and reproducibility of the methods. The literature review component could be pushed slightly further for more impact. While information on all REM modelling papers is collected, the results and discussion focus only on the presentation of the REM papers that consider predator-prey systems. It’d be great to see an assessment of the REM assumption violation across all REM modelling literature- rather than just the focus on the small subset. Overall though a great read, thanks for the opportunity to review.
Response: Thank you for providing a helpful review that has improved our manuscript.
Line 18: Arguably all species do not use landscapes randomly- perhaps delete ‘predator’ here.
Response: We agree, but because this study and manuscript are focused on the predator-prey framework, we have retained ‘predator’ and added “and other species”.
Line 30: A bit of detail lacking here for the uninitiated. Perhaps define the issue more clearly by changing to: ‘… lack individual unique natural markings required to undertake standard density estimation analyses’.
Response: We revised this sentence to reflect the reviewer’s suggestion.
Line 63: ref?
Response: We added references for this statement.
Line 77: The terminology is a little tricky here. Mark resight methods are used in camera trapping studies where the animals are individually identifiable, in that individuals ‘marked’ when their patterns are first seen on camera and ‘resighted’ when the patterns are reseen on subsequent images. Also, physical capture/marking methods were developed prior to the widespread use of camera trapping. Perhaps change this sentence to: Approaches have been used to resolve this individual…’ and line 84 to: ‘However such approaches…’.
Response: We disagree. The original impetus for developing mark-resight models was because most wildlife species do not have individually unique natural markings and therefore a portion of individuals in a population needed to be physically captured and uniquely marked for identification during subsequent resighting occasions that occurred via noninvasive methods that did not require physical recaptures (e.g., direct observations, camera-traps, etc.; see McClintock and White [2010; https://doi.org/10.1007/s10336-010-0524-x]). What the reviewer described (individuals are marked based on unique pelage patterns when first seen on camera and resighted on camera during later occasions) is noninvasive mark-recapture, not mark-resight. This is because in the reviewer’s described scenario, all individuals in a population naturally have individually unique natural markings, whereas in mark-resight only a portion of individuals have individually unique markings that were applied by researchers and all other individuals in the population are unmarked with unknown identities. In other words, mark-resight models incorporate detections of both marked and unmarked individuals, whereas mark-recapture models incorporate detections of only marked individuals. Our use of mark-resight in the manuscript is accurate and therefore we have not implemented the reviewer’s suggestion.
Line 85: I’m not sure this paragraph on indices is needed. Indices have their place, but they don’t generally provide estimates for density which is what is being tested/discussed here ... To tighten the intro and keep it flowing, I’d suggest deleting this para apart from the last sentence, and using that as the first sentence in the next paragraph.
Response: Although we understand the reviewer’s concern, we contend that the paragraph is warranted given the widespread use of indices to derive densities or abundances from camera-trap detections of unmarked wildlife. We have made it clear with this paragraph that indices do not produce estimates of those demographic parameters and that indices are typically unreliable surrogates for empirical, model-based estimates.
Line 105: I’d link these next two paragraphs together and whittle it down a bit if you can, as they really talk about the same thing. That’d produce a tighter 5 paragraph intro.
Response: We combined these two paragraphs and reduced the number of words.
Line 130: I think you could finish in the final part of this para with a statement about the unknowns that your study will answer. Currently the lit review ideas mentioned at line 141 are a bit out of the blue, here you could set the reader up to expect this. For example, add a sentence that states something along the lines of: ‘To date there have been few empirical tests of movement velocity and camera placement impacts on REM density estimates, and no assessment of the extent this is an issue in REM model implementation in the predator-prey literature’.
Response: Thank you for this suggestion. We added a sentence similar to the reviewer’s suggestion.
Line 202: Really nice methods.
Response: Thank you.
Line 389: State if non-english language papers were considered.
Response: We clarified in the revised manuscript that both English and non-English language papers were included in the literature review.
Line 402: I’m a little confused here about the model structure. What are the response and predictor variables in each model? Have you run three models each with a different response variable and the same predictors? Also describe the model selection process- have you just run the global model or run a model set? Perhaps a table in the supplementary material describing the models, variables included and the AIC values (ect) would help.
Response: We added details to this section in the Methods to clarify what the response and predictor variables were. Specifically, either the number of studies or number of density estimates were the response variable; an interaction between year and either camera placement or velocity source were the predictor variables. We clarified in the manuscript that AICc model selection was used to determine which model and therefore distribution was most supported for each response variable (i.e., a Poisson model and a negative-binomial model was fit for each response and predictor combination, and AICc was used to determine which model to produce parameter estimates from).
Line 534: I could be missing something here but to me it looks like the search was conducted for REM papers generally, but the results are focused on the presentation of REM studies that are focused on a predator-prey system. While this is relevant to the current study, it would be great to also have info on the broader REM literature and how misleading that is. This seems particularly important given the sample sizes involved (84 REM studies, but only 17 REM studies in a predator-prey framework) and with the literature collated there is a good opportunity to do this.
Response: Although we agree that these issues in the broader REM literature is important, we also agree with the reviewer that focusing on the predator-prey REM studies is the most relevant to our study and believe that focusing on the broader REM literature is somewhat beyond the scope of the study. However, we presented the results for the broader REM literature in Appendix A, and in the revised Discussion, we added some mention of the issues in the broader REM literature relative to the issues in the predator-prey REM literature.
Line 551: Again, slightly more info on modeling approach is needed- what was the model structure, and if there were competing models in the same model set, how much better was the best fitting model? Perhaps add a table even in the supplementary material to lay it out.
Response: We added a supplemental table (Table S1) that shows the model structures and model selection results for each analysis.
Line 572: There’s an opportunity with the first paragraph of the discussion to grab the readers attention and to tell them the main findings. This has been done well in the concluding paragraph, but would be useful to do here. As it stands the reader needs to wade through this paragraph before getting to any key findings or results.
Response: We revised the opening paragraph of the Discussion per the reviewer’s recommendation.
Line 593: This is a bit of a tangent, but I wonder for species like coyotes that use habitat in a very nonrandom way (choosing to move along features like tracks and roads) - how accurate are REM estimates likely to be even if random detectors arrangements are used? Eg are there species that this method isn’t appropriate for just because of their biology? I feel like the missing piece here is to compare density estimates between SECR and REM for species that use the landscape in a more or a less random way. This is just more of a thought bubble no need to change anything..
Response: We agree with the reviewer. To our knowledge, non-random space use by species like coyotes and other terrestrial carnivores has never been explicitly considered in previous comparisons between REM and SCR density estimates. However, some previous studies found the REM produced lower density estimates for carnivores than SCR, and authors posited that the REM densities were negatively biased for those species.
Line 606: It seems like there were also season differences in the extent this occurred, perhaps worth mentioning- up to you.
Response: Thank you. We added mention of pertinent seasonal differences that we observed.
Line 635: Here there is a discussion of the finding that most REM density estimates in predator prey literature are misleading- that’s super interesting. It would be great to also see a comparison and discussion of these figures generally (outside of predator and prey literature). Do the density estimates in the broader REM literature also rely on erroneous estimates of velocity and strategic camera placements?
Response: This information in the broader REM literature was included in Appendix A. However, we added some discussion in the Discussion section that covers the similarities and differences between predator-prey REM studies and the broader REM literature.
Round 2
Reviewer 1 Report
Comments and Suggestions for Authors
GENERAL COMMENTS
The addition of the limitations section, and some additional caveats in the discussion were effectively written and sufficient to address most of those related concerns. The addition of the supplemental figure showing tracking data is also a welcome addition and provides critical context for assessing the study design. The authors largely addressed additional concerns from my initial review, with just a few exceptions I detail below, though it was somewhat challenging to identify the exact changes made when entire paragraphs and sections were shown as tracked changes, instead of just those sections that were modified.
The key concerns that remain are related to the GLM models themselves, and the data going into them. See comments below.
Below I am using line numbers from the PDF document (which shows the tracked changes).
INTRODUCTION
Line 224: Should be “precludes”
Line 231: change to “…than at randomly placed camera-traps.”
Line 263: Remove comma after “…decisions based on…”
METHODS
Line 400: I suggest adding to the end or beginning of this sentence to put this in context. For example at the end could add “…to estimate key movement metrics for using tracking data from both species”. As it reads now, it’s not clear what the models are for and the paragraph theme isn’t well established.
Line 455: Here “…satellite location data from radio-collars…” seems a bit confusing. From what I know of the devices used, they would not generally be called “radio-collars”, a term typically reserved for VHF or UHF tracking devices. In addition GPS location data seems more common and clear than “satellite”. Presumably these devices use a satellite network to transmit the data to the researcher as well, so GPS is more clear here and elsewhere. Line 504 uses “GPS radio-collar”, which I would change to “GPS collar”. Check for consistency throughout.
Line 486: Change to “…whether or not they had radio-collars or ear tags.” Or you could use “regardless of the presence of radio-collars.”
Regarding the GLM models, which are indeed now better explained, I think the clarity could be further increased. I think you could explain that you have two response variables, and three predictors. You’ll use the results of year only model to identify overall temporal trends for each response. Importantly, this model should be compared to an intercept only model. If AIC favors the year model, then this is the evidence for a temporal trend. The ideal way to use AIC from there is to compare the year only model, with year * velocity method and then year only compared with year * camera placement. If the year only model is better in either case, then there is no evidence for additional patterns in relation to that predictor. This comparison with the base year only model is important, because it allows you to assess whether there is indeed evidence for an additional trend in your predictor, beyond a general temporal trend overall. Also please specify in this paragraph what the exact levels of each predictor are. They are explained earlier but I think it would be helpful to remind readers what the options are for velocity estimation and camera placement.
I also strongly suggest that the authors base the decision of Poisson vs. NB distribution on the overdispersion of the data itself. These two distributions are used to model count data, and their relative appropriateness is based on whether the data are overdispersed (and the associated behavior of residuals and their variation). I am not familiar with the use of AIC model selection to compare fit of identical models with these two distributions (note that the NB has an additional parameter and so AIC will be biased against that model from the start), and the authors would have to provide a reference for why this might be appropriate. From what I understand, because the relative appropriateness of these two distributions relates to the magnitude of expected change in the variation of the residuals as counts increase, it is not clear that AIC comparisons would be effective at identifying the most appropriate distribution. Given this, I strongly suggest simply using one of the available tests for overdispersion on the “full” model for each of the 4 analyses, then moving forward with the appropriate distribution in each case. I see from your supplemental table that one NB model did not converge. I suspect this may be due to the low sample size in that model, which relates to a related concern below in this model/analysis. But if these data are indeed overdispersed, and the NB model does not converge, then this regression approach is not defensible for that particular data set and question.
RESULTS
I have a follow up regarding this exchange from the first review:
“Line 413: Here you present average total number of locations collected within season/year combinations. This implies that the analysis of the movement data will pool locations across individuals. However in the paragraphs below it seems that movement velocities were estimated at an individual level and then averaged. If so, this should be clarified in the methods, and in this part of the results it would be more relevant to report the range of locations at an individual level, unless there are also some analyses that pool locations across individuals.”
“Response: Movement velocities were estimated at the individual-level and then averaged within each season × year combination, which we reported in the Methods section of the original manuscript (lines 349-352). Given the REM requires mean velocity values within the timeframe of camera sampling (e.g., season), we believe that it is more relevant to report the number of locations for each season.”
It was indeed clear from the original text what was done, but the reporting of total locations remains an issue here. If you are only reporting total locations for each season/year combination, this does not give a clear sense of the data going into your average movement velocities. For example, nearly all the locations could have potentially come from one animal, with 10-20 locations from each of the rest of the animals. I’m sure this isn’t the case but the results aren’t presented in a way to allow that to be clear. It looks like you do show a range of values, which displays the range across season/years. But I think what the reader is lacking is at least the range of locations, per season/year, for each species, that individual movement velocities were based on. I guess this gets back to the question of “sufficient monitoring” that the authors somewhat addressed in the following question from the first review. There again though I was more asking about whether there was a minimum number of locations used as a threshold for estimating movement velocities (or minimum effective sample size in ctmm). You could potentially just state the overall minimum # of locations used across all season years for any individual (for each species) if this value is relatively high. “Movement velocities were estimated in all cases using at least X and X tracking locations for coyotes and jackrabbits respectively”. Or, you could provide a supplemental table that lists all the movement velocities estimated, across all season/years and species, listing the sample sizes for each, and potentially even the models selected. I think the reader just needs a slightly better idea of what data went into the estimation of these movement velocities.
Regarding this response: “Finally, there was no time block during 2017-2023 that was separated in some analyses and that is not conveyed on any of the figures. We suspect the reviewer is referring to what appears to be missing data in the background columns for the year 2021 on Figure 6A and 6B, but that is simply the accurate reflection that no REM density estimates were produced in the predator-prey framework during 2021, which we correctly recorded in our literature review dataset as a 0 value for year 2021.”
To clarify, I was referring to this text which remains in the manuscript when I made this comment: “that rate differed during 2017–2023 by both camera placement and velocity value source, such that most densities in predator-prey ecology studies were produced using strategically placed cameras (p = 0.02–0.04; Figure 6C) and borrowed velocities (p = 0.004–0.03; Figure 6D).” I suppose then this reference to 2017-2023 is simply based on a visual assessment of the figures as opposed to some statistical assessment? You repeat this reference to 2017-2023 in the Appendix when referring to the full data set. The way it is written, with p-values at the end of the sentence, implies that this specific time period was tested statistically. So either remove, or clarify that this is based on visual assessment of the figure.
I’d like to follow up on this discussion from the first review:
Here is my original comment: “I would strongly suggest removing the analysis of each density estimate as a result. In addition, 14 is a very low sample size for the predator prey data set to investigate for temporal trends. I cannot think of a reason why this particular type of study would employ different REM methods on average than the REM literature in general. And the larger dataset is a much more defensible approach to looking at temporal trends. So I suggest only looking at trends over time in REM usage, using the full dataset (move Appendix results in some form into the full paper).”
And this was the response: “Further, as is evident by comparing the predator-prey REM study results (Figure 6) and the broader REM study results (Figure A1), the trends in inappropriate usage of camera placement types are different, whereas the trends in velocity value source are similar. Thus, the reviewer’s assumptions about predator-prey REM studies relative to the broader REM studies are unsupported and we have retained the analysis and results as presented in the original manuscript. “
I do not agree that there is clear evidence of robust differences between the predator-prey studies and the overall data. It is clear from the figures that patterns in those trends are driven largely by just a few data points (years) and this fact is partly what fueled my original concern. The total sample of studies focused on predator-prey data is much lower than the overall sample and therefore much more heavily influenced by just a few studies. From what I understand, you’ve taken a total of 14 predator-prey studies, and split this count across all the years. Most of the years have 0 or 1 study. At the very least, some mention of this limited sample size, and potentially an outlier analysis that removes points identified as outliers and reruns the model, should be included.
To address my concern about the lack of independence in the density estimate analysis the authors responded: “We disagree, primarily because multiple studies produced multiple density estimates for multiple species using both random and strategic camera placements and/or borrowed and study-specific velocity values, which varied by species and study area within a given study. This can be seen in the supplementary Spreadsheet S1 of the literature review results that we provided. For example, Caravaggi et al. (2016) produced 5 density estimates for two species, one species of which they obtained velocities for using data from the study area, the other species of which they borrowed velocities for from a separate study in a different area. Similarly, Rahman et al. (2017) produced 6 density estimates for one species, but 2 estimates were from strategically placed cameras, 2 estimates were from randomly placed cameras, and 2 estimates were from both strategically and randomly placed cameras. Thus, by focusing on the study alone, instead of also the density estimates, such discrepancies would be lost and paint an inaccurate picture of what has occurred.”
Although it is true that some studies present estimates based on variable methods, there are also a sizable number of examples that show clearly that all density estimates are not independent from a statistical standpoint, and these were not mentioned by the authors in their response. For example, there is a Mengulluoglu study with 9 estimates and all use strategic camera placement…a Yang study has 9 estimates, all of which are based on strategic camera placement and borrowed velocities…a Cardoso study has 14 estimates, all of which are based on strategic and borrowed. There are many other similar examples of this. This shows clearly that each density estimate is not an independent sample of how common these methods are in that year and therefore they cannot be treated as independent samples of this trend. Instead of removing this analysis, I believe this could be resolved by switching to a mixed effects model, and specifying a random intercept for study ID. But as currently presented, this GLM analysis is not appropriate for the density estimate response variable.
TABLE S1 is helpful. In the revision it should include only one distribution (Poisson or NB) for each scenario, but should also include the intercept only model, and for a given response variable, all 4 models can be listed together, in AIC order (intercept, year, year * camera placement, year * velocity method.
Author Response
Comment 1: The addition of the limitations section, and some additional caveats in the discussion were effectively written and sufficient to address most of those related concerns. The addition of the supplemental figure showing tracking data is also a welcome addition and provides critical context for assessing the study design. The authors largely addressed additional concerns from my initial review, with just a few exceptions I detail below, though it was somewhat challenging to identify the exact changes made when entire paragraphs and sections were shown as tracked changes, instead of just those sections that were modified. The key concerns that remain are related to the GLM models themselves, and the data going into them. See comments below. Below I am using line numbers from the PDF document (which shows the tracked changes).
Response 1: Thank you for providing another helpful review. We are unsure why the reviewer only had access to the track changes version of the revised manuscript, because we uploaded both a clean, non-track changes version and a track changes version, as required by the journal.
Comment 2: Line 224: Should be “precludes”
Response 2: We changed this to ‘precludes’.
Comment 3: Line 231: change to “…than at randomly placed camera-traps.”
Response 3: We made this change.
Comment 4: Line 263: Remove comma after “…decisions based on…”
Response 4: We removed the comma.
Comment 5: Line 400: I suggest adding to the end or beginning of this sentence to put this in context. For example at the end could add “…to estimate key movement metrics for using tracking data from both species”. As it reads now, it’s not clear what the models are for and the paragraph theme isn’t well established.
Response 5: We revised the beginning of the paragraph to, “To estimate movement velocities for both species, we fit…”
Comment 6: Line 455: Here “…satellite location data from radio-collars…” seems a bit confusing. From what I know of the devices used, they would not generally be called “radio-collars”, a term typically reserved for VHF or UHF tracking devices. In addition GPS location data seems more common and clear than “satellite”. Presumably these devices use a satellite network to transmit the data to the researcher as well, so GPS is more clear here and elsewhere. Line 504 uses “GPS radio-collar”, which I would change to “GPS collar”. Check for consistency throughout.
Response 6: We changed all mentions of ‘radio-collar’ to ‘GPS collar’ throughout the manuscript.
Comment 7: Line 486: Change to “…whether or not they had radio-collars or ear tags.” Or you could use “regardless of the presence of radio-collars.”
Response 7: We changed this to, “…regardless of whether or not they had GPS collars or ear tags.”
Comment 8: Regarding the GLM models, which are indeed now better explained, I think the clarity could be further increased. I think you could explain that you have two response variables, and three predictors. You’ll use the results of year only model to identify overall temporal trends for each response. Importantly, this model should be compared to an intercept only model. If AIC favors the year model, then this is the evidence for a temporal trend. The ideal way to use AIC from there is to compare the year only model, with year * velocity method and then year only compared with year * camera placement. If the year only model is better in either case, then there is no evidence for additional patterns in relation to that predictor. This comparison with the base year only model is important, because it allows you to assess whether there is indeed evidence for an additional trend in your predictor, beyond a general temporal trend overall. Also please specify in this paragraph what the exact levels of each predictor are. They are explained earlier but I think it would be helpful to remind readers what the options are for velocity estimation and camera placement.
Response 8: We disagree with the reviewer. As noted in the Literature Review subsection of the Methods (lines 410-422 in the non-track changes revision), we had clearly defined objectives for the GLM analysis: 1) quantify the rate of change over time in the number of studies that used the REM; 2) quantify the rate of change over time in the number of REM density estimates; 3) test whether the rate of change over time in the number of REM density estimates differed between camera placement strategy and velocity value source. The objective of the GLM analysis was not to find the overall best-fitting model among said candidate models to identify variables that are important for predicting the number of REM density estimates across time, which is what the reviewer has described. Instead, the entire purpose of the GLM analysis was to address the above specific objectives regarding the number of REM density estimates and the data sources that studies used to produce those estimates, for which an intercept-only model would be useless. As such, we have not revised our approach to include the reviewer’s modeling suggestions in this comment. However, we did clarify in the paragraph that there were two response variables and three total predictor variables, along with specifying the range or levels of each predictor.
Comment 9: I also strongly suggest that the authors base the decision of Poisson vs. NB distribution on the overdispersion of the data itself. These two distributions are used to model count data, and their relative appropriateness is based on whether the data are overdispersed (and the associated behavior of residuals and their variation). I am not familiar with the use of AIC model selection to compare fit of identical models with these two distributions (note that the NB has an additional parameter and so AIC will be biased against that model from the start), and the authors would have to provide a reference for why this might be appropriate. From what I understand, because the relative appropriateness of these two distributions relates to the magnitude of expected change in the variation of the residuals as counts increase, it is not clear that AIC comparisons would be effective at identifying the most appropriate distribution. Given this, I strongly suggest simply using one of the available tests for overdispersion on the “full” model for each of the 4 analyses, then moving forward with the appropriate distribution in each case. I see from your supplemental table that one NB model did not converge. I suspect this may be due to the low sample size in that model, which relates to a related concern below in this model/analysis. But if these data are indeed overdispersed, and the NB model does not converge, then this regression approach is not defensible for that particular data set and question.
Response 9: We strongly disagree with the reviewer. Numerous statistical texts have detailed the appropriateness of using AIC- and, alternatively, BIC-based model selection for determining whether a Poisson or negative-binomial model (among other GLM distributions) is a better fit to the data (e.g., Hilbe [2014; https://www.cambridge.org/9781107611252], Harris et al. [2014; https://doi.org/10.1177/1536867X1401400306], Hardin and Hilbe [2018; https://www.stata.com/bookstore/generalized-linear-models-and-extensions/]). In short, both the Poisson and NB are full discrete distributions, resulting in the likelihoods being full, and AIC appropriately penalizes NB models for the extra parameter (Lindsey and Jones 1998; https://doi.org/10.1002/(SICI)1097-0258(19980115)17:1%3C59::AID-SIM733%3E3.0.CO;2-7). This would not be the case with, e.g., quasi-Poisson models, because they do not have a full distributional form and therefore cannot be compared with full-distributional models, such as Poisson or NB models, via AIC (see Burnham and Anderson [2002]; VerHoef and Boveng [2007; https://www.jstor.org/stable/27651434]). Furthermore, the glmmTMB package that we used to fit the models computes the full likelihoods of both Poisson and NB models, allowing their direct comparison via AIC (Brooks et al. 2017; https://doi.org/10.32614/RJ-2017-066; note that paper also uses AIC to select among Poisson and NB models).
Comment 10: I have a follow up regarding this exchange from the first review:
“Line 413: Here you present average total number of locations collected within season/year combinations. This implies that the analysis of the movement data will pool locations across individuals. However in the paragraphs below it seems that movement velocities were estimated at an individual level and then averaged. If so, this should be clarified in the methods, and in this part of the results it would be more relevant to report the range of locations at an individual level, unless there are also some analyses that pool locations across individuals.”
“Response: Movement velocities were estimated at the individual-level and then averaged within each season × year combination, which we reported in the Methods section of the original manuscript (lines 349-352). Given the REM requires mean velocity values within the timeframe of camera sampling (e.g., season), we believe that it is more relevant to report the number of locations for each season.”
It was indeed clear from the original text what was done, but the reporting of total locations remains an issue here. If you are only reporting total locations for each season/year combination, this does not give a clear sense of the data going into your average movement velocities. For example, nearly all the locations could have potentially come from one animal, with 10-20 locations from each of the rest of the animals. I’m sure this isn’t the case but the results aren’t presented in a way to allow that to be clear. It looks like you do show a range of values, which displays the range across season/years. But I think what the reader is lacking is at least the range of locations, per season/year, for each species, that individual movement velocities were based on. I guess this gets back to the question of “sufficient monitoring” that the authors somewhat addressed in the following question from the first review. There again though I was more asking about whether there was a minimum number of locations used as a threshold for estimating movement velocities (or minimum effective sample size in ctmm). You could potentially just state the overall minimum # of locations used across all season years for any individual (for each species) if this value is relatively high. “Movement velocities were estimated in all cases using at least X and X tracking locations for coyotes and jackrabbits respectively”. Or, you could provide a supplemental table that lists all the movement velocities estimated, across all season/years and species, listing the sample sizes for each, and potentially even the models selected. I think the reader just needs a slightly better idea of what data went into the estimation of these movement velocities.
Response 10: Thank you for the clarification about your original comment. Admittedly, we misunderstood the comment. We fully understand now and have revised these parts of the manuscript. We added the number of locations/individual/season/year for each species to lines 449 and 464 (non-track changes revised manuscript), and we added that seasonal movement velocities within each year were estimated using a minimum of 200 and 100 locations/individual for coyotes and jackrabbits, respectively, to lines 266-267 (non-track changes revised manuscript).
Comment 11: Regarding this response: “Finally, there was no time block during 2017-2023 that was separated in some analyses and that is not conveyed on any of the figures. We suspect the reviewer is referring to what appears to be missing data in the background columns for the year 2021 on Figure 6A and 6B, but that is simply the accurate reflection that no REM density estimates were produced in the predator-prey framework during 2021, which we correctly recorded in our literature review dataset as a 0 value for year 2021.”
To clarify, I was referring to this text which remains in the manuscript when I made this comment: “that rate differed during 2017–2023 by both camera placement and velocity value source, such that most densities in predator-prey ecology studies were produced using strategically placed cameras (p = 0.02–0.04; Figure 6C) and borrowed velocities (p = 0.004–0.03; Figure 6D).” I suppose then this reference to 2017-2023 is simply based on a visual assessment of the figures as opposed to some statistical assessment? You repeat this reference to 2017-2023 in the Appendix when referring to the full data set. The way it is written, with p-values at the end of the sentence, implies that this specific time period was tested statistically. So either remove, or clarify that this is based on visual assessment of the figure.
Response 11: This was not based on a visual assessment. As noted at the end of the Literature Review subsection of the Methods (lines 424-428 in the non-track changes revision), the statement and corresponding p-values are based on post-hoc Tukey’s multiple comparisons tests that were applied to the fitted model.
Comment 12: I’d like to follow up on this discussion from the first review:
Here is my original comment: “I would strongly suggest removing the analysis of each density estimate as a result. In addition, 14 is a very low sample size for the predator prey data set to investigate for temporal trends. I cannot think of a reason why this particular type of study would employ different REM methods on average than the REM literature in general. And the larger dataset is a much more defensible approach to looking at temporal trends. So I suggest only looking at trends over time in REM usage, using the full dataset (move Appendix results in some form into the full paper).”
And this was the response: “Further, as is evident by comparing the predator-prey REM study results (Figure 6) and the broader REM study results (Figure A1), the trends in inappropriate usage of camera placement types are different, whereas the trends in velocity value source are similar. Thus, the reviewer’s assumptions about predator-prey REM studies relative to the broader REM studies are unsupported and we have retained the analysis and results as presented in the original manuscript. “
I do not agree that there is clear evidence of robust differences between the predator-prey studies and the overall data. It is clear from the figures that patterns in those trends are driven largely by just a few data points (years) and this fact is partly what fueled my original concern. The total sample of studies focused on predator-prey data is much lower than the overall sample and therefore much more heavily influenced by just a few studies. From what I understand, you’ve taken a total of 14 predator-prey studies, and split this count across all the years. Most of the years have 0 or 1 study. At the very least, some mention of this limited sample size, and potentially an outlier analysis that removes points identified as outliers and reruns the model, should be included.
Response 12: We added mention of the small sample size of predator-prey REM studies relative to the broader REM literature to the Discussion section on lines 649-650 (non-track changes revised manuscript). However, we did not conduct an outlier analysis or consider removing outliers, because none of the valid reasons for removing outliers were present in our literature review data (e.g., measurement/sampling error, data entry error, not a part of the population under study). Removing any yearly counts of REM studies or REM density estimates due to appearing as an outlier would artificially reduce the variation inherent to these counts and result in a fitted model that makes the counts appear more predictable than they actually are in reality, which is poor statistical practice.
Comment 13: To address my concern about the lack of independence in the density estimate analysis the authors responded: “We disagree, primarily because multiple studies produced multiple density estimates for multiple species using both random and strategic camera placements and/or borrowed and study-specific velocity values, which varied by species and study area within a given study. This can be seen in the supplementary Spreadsheet S1 of the literature review results that we provided. For example, Caravaggi et al. (2016) produced 5 density estimates for two species, one species of which they obtained velocities for using data from the study area, the other species of which they borrowed velocities for from a separate study in a different area. Similarly, Rahman et al. (2017) produced 6 density estimates for one species, but 2 estimates were from strategically placed cameras, 2 estimates were from randomly placed cameras, and 2 estimates were from both strategically and randomly placed cameras. Thus, by focusing on the study alone, instead of also the density estimates, such discrepancies would be lost and paint an inaccurate picture of what has occurred.”
Although it is true that some studies present estimates based on variable methods, there are also a sizable number of examples that show clearly that all density estimates are not independent from a statistical standpoint, and these were not mentioned by the authors in their response. For example, there is a Mengulluoglu study with 9 estimates and all use strategic camera placement…a Yang study has 9 estimates, all of which are based on strategic camera placement and borrowed velocities…a Cardoso study has 14 estimates, all of which are based on strategic and borrowed. There are many other similar examples of this. This shows clearly that each density estimate is not an independent sample of how common these methods are in that year and therefore they cannot be treated as independent samples of this trend. Instead of removing this analysis, I believe this could be resolved by switching to a mixed effects model, and specifying a random intercept for study ID. But as currently presented, this GLM analysis is not appropriate for the density estimate response variable.
Response 13: The Poisson and negative-binomial assumption of independence applies specifically to the counts across a finite observation space, which, in the case of our literature review data, is years. Thus, the independence assumption translates to: 1) the counts of density estimates within a given year must be independent from the counts of density estimates in other years, and 2) the counts from each study within a year must be independent from the counts from other studies within that same year. Our data satisfy both of those independence requirements, because: 1) it is impossible for a given study to be published in more than one year (independence of estimate counts among studies among years satisfied), and 2) there is no reason to believe that two studies published in the same year were dependent on each other (independence of estimate counts among studies within a year satisfied). Therefore, adding random intercepts for studies in a mixed effects model is unwarranted and unnecessary. The simplest example of when a mixed effects Poisson or NB model should be used is in longitudinal studies that obtain repeated observations on the same individuals or the same study areas across time; for instance, counts of field mice are recorded in the same 10 agriculture fields across each of 10 days, which would necessitate group-level random intercepts for the 10 fields because the counts were recorded in each field every day, so the daily counts within a given field are not independent.
Comment 14: TABLE S1 is helpful. In the revision it should include only one distribution (Poisson or NB) for each scenario, but should also include the intercept only model, and for a given response variable, all 4 models can be listed together, in AIC order (intercept, year, year * camera placement, year * velocity method.
Response 14: Again, we strongly disagree with the reviewer’s suggestion, primarily because we had defined objectives for the GLM analysis that we described in the Methods to be addressed. The purpose of the GLM analysis was not to find the best fitting model, among a suite of candidate models, of the most important variables that influence the number of REM studies or REM density estimates across time. Our objectives were to: 1) quantify the rate of change over time in the number of studies that used the REM; 2) quantify the rate of change over time in the number of REM density estimates; 3) test whether the rate of change over time in the number of REM density estimates differed between camera placement strategy and velocity value source. An intercept-only model is useless for addressing those objectives. Our sole purpose of using AIC for the GLM analysis was to determine if the Poisson or NB variant of each model specification was the better fit to the data for addressing each objective.